# Hrq1/RECQL4 regulation is critical for preventing aberrant recombination during DNA intrastrand crosslink repair and is upregulated in breast cancer

Thong T. Luong[1], Zheqi Li[2], Nolan Priedigkeit[2], Phoebe S. Parker[1], Stefanie Böhm[1], Kyle Rapchak[1], Adrian V. Lee[2], Kara A. Bernstein[1]¤*

1 University of Pittsburgh School of Medicine, Department of Pharmacology and Chemical Biology, UPMC Hillman Cancer Center, Pittsburgh, Pennsylvania, United States of America, 2 Women's Cancer Research Center, UPMC Hillman Cancer Center, Magee-Womens Research Institute, Pittsburgh, Pennsylvania, United States of America

¤ Current address: University of Pennsylvania School of Medicine, Department of Biochemistry and Biophysics, Philadelphia, Pennsylvania, United States of America
* kara.bernstein@pennmedicine.upenn.edu

**Data Availability Statement:** All relevant data are within the manuscript and its Supporting Information files.

## Abstract

Human RECQL4 is a member of the RecQ family of DNA helicases and functions during DNA replication and repair. *RECQL4* mutations are associated with developmental defects and cancer. Although *RECQL4* mutations lead to disease, *RECQL4* overexpression is also observed in cancer, including breast and prostate. Thus, tight regulation of RECQL4 protein levels is crucial for genome stability. Because mammalian *RECQL4* is essential, how cells regulate RECQL4 protein levels is largely unknown. Utilizing budding yeast, we investigated the *RECQL4* homolog, *HRQ1*, during DNA crosslink repair. We find that Hrq1 functions in the error-free template switching pathway to mediate DNA intrastrand crosslink repair. Although Hrq1 mediates repair of cisplatin-induced lesions, it is paradoxically degraded by the proteasome following cisplatin treatment. By identifying the targeted lysine residues, we show that preventing Hrq1 degradation results in increased recombination and mutagenesis. Like yeast, human RECQL4 is similarly degraded upon exposure to crosslinking agents. Furthermore, over-expression of *RECQL4* results in increased RAD51 foci, which is dependent on its helicase activity. Using bioinformatic analysis, we observe that *RECQL4* overexpression correlates with increased recombination and mutations. Overall, our study uncovers a role for Hrq1/RECQL4 in DNA intrastrand crosslink repair and provides further insight how misregulation of RECQL4 can promote genomic instability, a cancer hallmark.

## Author summary

RECQL4 is a DNA helicase and functions during DNA replication and repair. While loss-of-function RECQL4 mutations are found in diseases characterized by developmental defects and cancer, such as Rothmund-Thomson syndrome, over-expression of RECQL4

**Funding:** This study was supported by grants from the National Institutes of Health (ES030335 to K.A. B.) and the Department of Defense (BC201356 to K.A.B.). This work was also supported by the Hillman Fellows for Innovative Cancer Research Program to K.A.B. The funders had no role in study design, data collection and analysis, decision to publish, or preparation of the manuscript.

**Competing interests:** The authors have declared that no competing interests exist.

is also observed in cancer, such as breast cancer. Therefore, RECQL4 protein expression must be tightly regulated. Here we used the budding yeast homolog of RECQL4, Hrq1, and discovered that overexpression of Hrq1 protein levels result in increased recombination and mutations, both cancer hallmarks. We find that Hrq1 functions to mediate repair of a specific type of DNA damage, intrastrand crosslinks, which occur when DNA nucleotides on the same strand are chemically linked together. These findings are also conserved in humans suggesting a common mechanism between yeast Hrq1 and human RECQL4. Overall, our study identifies a conserved role for RECQL4 in DNA intrastrand crosslink repair and provides insights into how its misregulation could promote cancer development.

## Introduction

Accurate and timely repair of DNA damage is critical for genomic integrity and human health. Disruptions of DNA repair genes are frequently associated with diseases such as cancer and aging. One such gene is *RECQL4*, which belongs to the evolutionarily conserved family of RecQ helicases. This family of 3' to 5'- DNA helicases is often referred to as the "Guardians of the Genome" due to crucial roles in DNA recombination, replication, and repair that are conserved from yeast to man [1–5]. Mutations in *RECQL4* are associated with three heritable autosomal diseases: Rothmund-Thomson syndrome (RTS type II), Baller-Gerold syndrome (BGS), and RAPADILINO, each characterized by developmental defects, cancer and/or premature aging [6–12]. Although *RECQL4* dysfunction is associated with hereditable diseases, recent studies have shown that overexpression of *RECQL4* is linked to multiple cancer types such as breast, hepatic, gastric, and prostate. In each cancer type, high levels of *RECQL4* are correlated with poor prognosis [13–16]. Since both inactivation or overexpression of *RECQL4* cause human disease and genetic instability, *RECQL4* protein levels must be tightly regulated.

Although *RECQL4* is critical for genome integrity and disease prevention, previous studies of mammalian *RECQL4* were stymied due to technical difficulties, including the embryonic lethality of mouse knockout models and the inviability of human *RECQL4* knockout cell lines [8,17–19]. In 2008, *HRQ1*, was discovered as the *Saccharomyces cerevisiae* homolog of *RECQL4* [20]. Since *HRQ1* is non-essential, yeast is a valuable model to elucidate *RECQL4* gene family function. For example, analysis of *hrq1Δ* cells identified a novel function for Hrq1 in DNA crosslink repair [21,22]. *HRQ1*-disrupted cells are sensitive to DNA crosslinking agents such as cisplatin and mitomycin C (MMC), which predominantly create intrastrand or interstrand crosslinks, respectively. Furthermore, recent studies suggest that human *RECQL4* also has a role in crosslink repair since *RECQL4* shRNA knockdown leads to cisplatin sensitivity in a triple-negative breast cancer (TNBC) cell line [14]. Although the *RECQL4* gene family has a conserved role in crosslink repair, its molecular function during this process has yet to be elucidated.

DNA crosslinking agents can induce two types of damage including, interstrand crosslinks (ICL) and intrastrand crosslinks. In ICLs, the Watson and Crick strands are covalently linked, whereas in an intrastrand crosslink, the same DNA strand is covalently linked to itself [23–26]. Due to the different nature of the crosslinks, the mechanism of how these two adducts are repaired is also unique. While there are some key differences in how ICLs are repaired from yeast to man, during replication-coupled ICL repair, the steps are largely conserved [27–31]. Following damage recognition, endonucleases nick either side of the damaged DNA, which then mediates exonuclease to come in and degrade the damaged DNA. This "unhooking" step results in a ssDNA gap. This gap may be filled in by the post-replicative repair (PRR) pathway

[28,32–35]. The PRR pathway consists of an error-prone and error-free branch. Utilization of the different branches is mediated by PCNA (Pol30 in yeast) ubiquitylation. For example, in both yeast and humans mono-ubiquitylation of PCNA at lysine 164 (K164) recruits error-prone translesion synthesis polymerases to fill the gap [36]. On the other hand, polyubiquitylation of the same K164 residue on PCNA results in error-free homology-directed repair [37]. In contrast to ICLs, intrastrand crosslinks are primarily repaired by the nucleotide excision repair (NER) pathway [38–41]. However, if the replication fork encounters the intrastrand crosslink and stalls, then the PRR and DNA damage tolerance pathways can mediate bypass of this adduct using the mechanism described above [36,37,42,43]. Subsequently NER, will excise and degrade the damaged DNA and the gap with be filled using DNA polymerases and ligases.

Recent studies demonstrate that Hrq1 functions during NER to repair cisplatin-induced DNA lesions [44–47]. However, there are conflicting studies as to whether Hrq1 also has a role in PRR [44–46]. For example, a study in *A. thaliana* suggests that Hrq1 functions in PRR while two independent studies in *S. pombe* and *S. cerevisiae*, respectively, suggest that Hrq1 functions independently of PRR [44–46]. Therefore, it remains controversial as to whether Hrq1 or its mammalian homology, RECQL4, truly functions during PRR.

Here, similar to plants and in contrast to previous studies in budding and fission yeast, we find that the budding yeast Hrq1 has a role in the error-free branch of PRR specifically during intrastrand crosslink repair. Furthermore, we performed genetic analysis that suggests that human *RECQL4* also functions during PRR. Paradoxically, despite a conserved role in cisplatin resistance, Hrq1 is degraded by the proteasome following cisplatin exposure. We find that *HRQ1* overexpression or its stabilization leads to increased recombination and mutation rates. Furthermore, suggesting a conserved regulatory mechanism, we show that endogenous human RECQL4 protein levels decrease following cisplatin and acetaldehyde treatment. Moreover, overexpression of *RECQL4* results in increased RAD51 foci, which is in part dependent on its helicase activity. Lastly, bioinformatic analysis reveals that high levels of *RECQL4* are correlated with increased tumor mutation burden. Importantly, we observe that in TNBC, the protein levels of RECQL4 are a predictive marker to cisplatin response. Our work uncovers a role for Hrq1 in the error-free branch PRR repair and a conserved regulatory mechanism between yeast Hrq1 and mammalian RECQL4 following DNA intrastrand crosslink damage.

## Results

### Hrq1 is degraded by the proteasome upon cisplatin treatment

Previous reports demonstrate that Hrq1 is important during DNA crosslink repair [21,22]. Consistent with prior studies [21,22], we observe that *hrq1Δ* cells are cisplatin sensitive compared to wild-type (WT) cells (**Fig 1A**). Since Hrq1 is needed for resistance to DNA crosslinking agents, we reasoned that Hrq1 protein levels may increase upon cisplatin exposure. To address this, we 9xMyc tagged Hrq1 on its C-terminus at its endogenous locus and promoter. We verified that 9xMyc-Hrq1 was functional upon exposure of the *HRQ1-9myc* yeast strain to cisplatin by comparing its growth to the parental WT strain (**S1A Fig**). Although Hrq1 is needed for cisplatin resistance, we paradoxically observe that Hrq1 steady state levels significantly decrease upon cisplatin exposure (**Fig 1B**). Importantly, Hrq1 degradation is specific to cisplatin since other types of DNA damage, such as methyl methanesulfonate (MMS; an alkylating agent), ionizing radiation (IR; which induces double-strand breaks), or hydroxyurea (HU; which depletes dNTP pools), have no detectible effect on Hrq1 steady state levels (**Fig 1B**). To verify these results, we repeated these experiments with multiple doses of MMS, IR, and HU and we observe that Hrq1 protein levels still remained unchanged (**S1B Fig**). These results suggest that the reduced Hrq1 protein levels observed are specific to cisplatin.

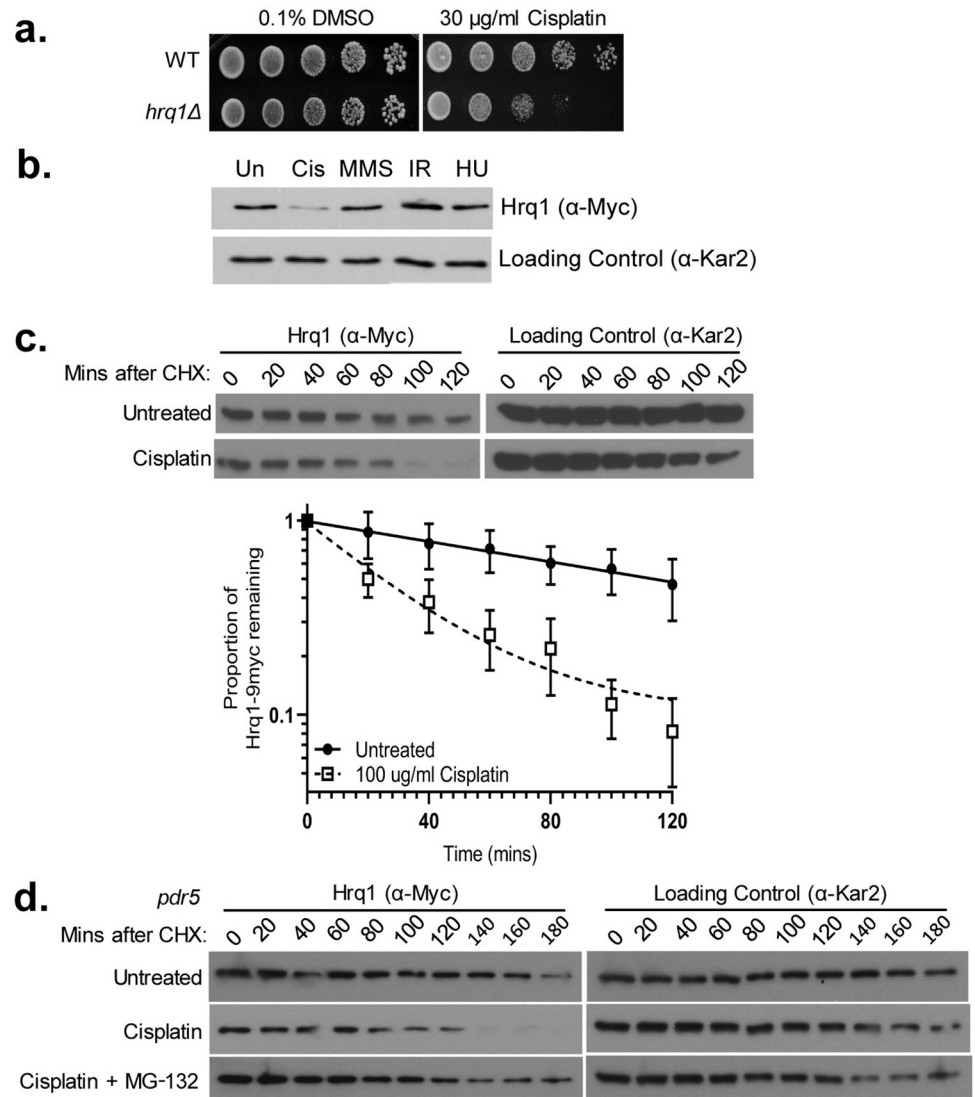

**Fig 1. Although Hrq1 is needed for cisplatin resistance, it is degraded by the proteasome upon cisplatin exposure.**
(A) *HRQ1*-null cells are sensitive to cisplatin. Wild-type (WT) or *hrq1Δ* disrupted cells were five-fold serially diluted
on medium containing 30 μg/ml cisplatin and/or 0.1% DMSO, grown for 48 hours at 30˚C, and photographed. (B)
Hrq1 level is stable following treatment with other DNA damaging agents: cisplatin (100 ug/ml), MMS (0.03%), IR
(100 Gy), HU (100 mM). Exponentially growing cells with Hrq1-9xMYC were treated with the indicated drugs for 2
hours before being harvested for western. (C) Hrq1 protein levels are decreased upon cisplatin treatment.
Exponentially growing cells with Hrq1-9xMYC were incubated with cycloheximide in the presence or absence of
100 μg/ml cisplatin and/or 0.1% DMSO. Quantification of the proportion of Hrq1 remaining relative to time 0 (before
CHX addition) and the loading control, Kar2. The experiment was performed five times with mean and standard error
plotted (Raw densitometry data in Sheets A-E in **S1 Data**). It is important to note that Hrq1 and the loading control,
Kar2, were analyzed from the same gels to account for pipetting errors. Since Hrq1 is not as abundant as the loading
control, there is a limitation for the densitometry analysis. (D) The proteasome degrades Hrq1 following cisplatin
exposure. *PDR5* disrupted cells were untreated (0.1% DMSO), cisplatin treated, or pretreated for one hour with 50 μM
MG-132 before cisplatin addition with 0.1% DMSO. Cycloheximide chases were performed the similarly as (B) but
further timepoints were taken.

Next, we sought to determine if the reduced Hrq1 protein levels observed in cisplatin-
treated cells are due to protein degradation. To measure Hrq1 protein stability, we performed
cycloheximide chase experiments and analyzed cells for Hrq1 protein every 20 minutes for 120

minutes following cycloheximide addition (**Fig 1C**). Consistent with our previous observations, cisplatin treatment led to a decrease in Hrq1 protein levels, where approximately 50% of Hrq1 remains 30 minutes following cycloheximide addition, whereas in DMSO treatment more than 50% of Hrq1 still remains at 120-minute mark (**Fig 1C**).

Proteasomal degradation is an important regulatory mechanism to ensure proper and timely DNA repair for many DNA repair proteins [48]. Therefore, one possibility is that Hrq1 protein levels are regulated by the 26S ubiquitin proteasome system (UPS) following DNA damage. To determine whether Hrq1 is degraded by the UPS following cisplatin treatment, we performed cycloheximide chase experiments in the presence of the proteasome inhibitor, MG-132 (**Fig 1D**). Hrq1 protein levels were stabilized following cisplatin exposure when the UPS is inhibited (**Fig 1D**). Note that in order to keep the intracellular concentration of MG-132 high, the drug efflux pump, *PDR5*, is also disrupted [49]. These results suggest that Hrq1 is marked for proteasomal degradation.

### The E3 ubiquitin ligase, Rad16, target Hrq1 for degradation following cisplatin

Since Hrq1 protein levels may be proteasome regulated, we sought to identify the E3 ubiquitin ligase that targets Hrq1. In *S. cerevisiae*, there are 60–100 E3 enzymes, so we prioritized E3 enzymes known to regulate DNA damage response proteins. For example, the NER gene, Rad4 (mammalian XPC), functions upstream of Hrq1 during crosslink repair, and its protein levels are regulated by the UPS following DNA damage [21,50]. Therefore, one possibility is that Rad4 and Hrq1 are targeted by the same E3 enzyme, Rad16 [51,52]. Thus, we examined whether Rad16 regulates Hrq1 protein levels. Indeed, we find that Hrq1 protein levels are largely stabilized in *rad16Δ* cells in cisplatin treated conditions (**Fig 2A**). Therefore, loss of *RAD16* E3 ubiquitin ligase results in Hrq1 protein stabilization following cisplatin exposure. Note that although deletion of *RAD16* stabilizes Hrq1 following cisplatin exposure, in untreated conditions Hrq1 levels still decrease. Therefore, there are likely additional E3 ubiquitin ligases that regulate Hrq1 protein levels independent of its role in the DNA damage response.

We also examined potential E2 enzymes that regulate Hrq1. In yeast, there are only thirteen known E2s. Of the E2 enzymes only two, Ubc13 and Rad6, are associated with the DNA damage response [53]. To test whether either of these two genes may regulate Hrq1, we knocked out either *RAD6* or *UBC13* in a Hrq1-9myc tagged strain and performed cycloheximide chase experiments. Disruption of either *UBC13* or *RAD6* only led to a mild stabilization of Hrq1 (**S2A and S2B Fig**). It is possible that in the absence of either *UBC13* or *RAD6*, that the other E2 enzyme could compensate for each other. To test this hypothesis, we created a double *ubc13Δ rad6Δ* knockout and analyzed Hrq1 protein levels by cycloheximide chase. We observe that deletion of both *UBC13* and *RAD6* does not fully stabilize Hrq1 protein levels (**S2C Fig**). These results suggest *UBC13* and *RAD6* do not compensate for each other and there are likely additional E2s that regulate Hrq1 protein levels.

### Hrq1 functions during error-free PRR

As stated above, DNA crosslinks are repaired using different pathways depending upon the cell cycle stage. For example, during replication, DNA intrastrand crosslinks are bypassed by the PRR pathway and then subsequently repaired by NER. Therefore, to determine whether Hrq1 functions during a specific cell cycle stage, we asked whether its protein levels are cell cycle regulated. We synchronized Hrq1-9xMYC expressing cells in G1 with alpha factor and released them into fresh medium to enable cell cycle progression. Protein extracts were made

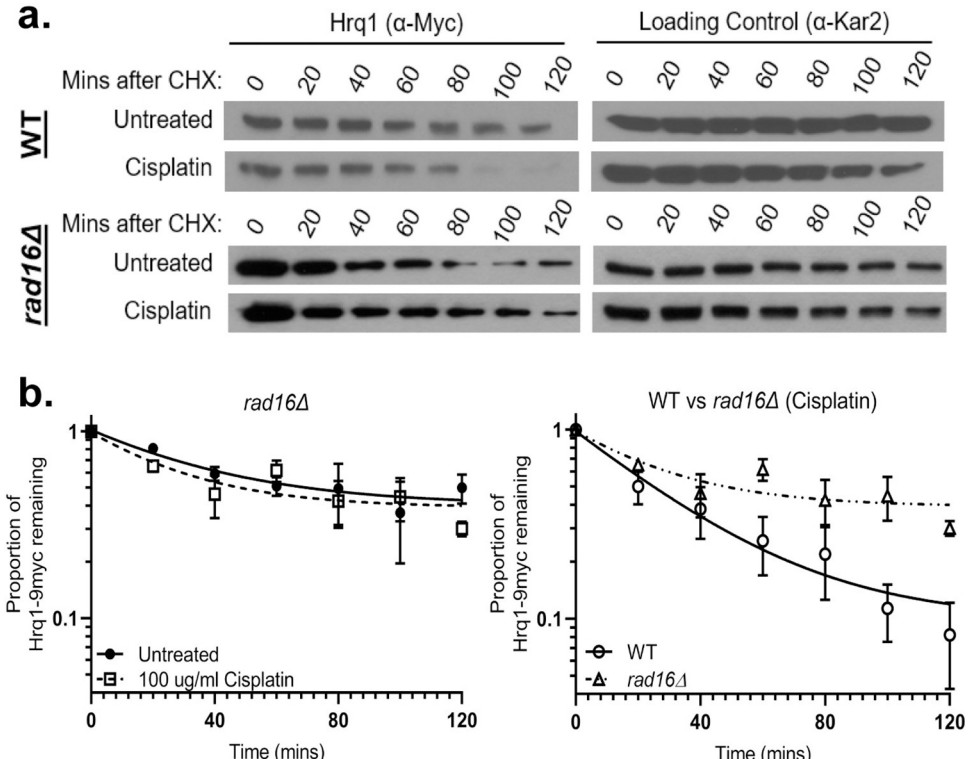

**Fig 2. Hrq1 protein levels are stabilized in the absence of the E3 Ub-ligase, *RAD16*.** (A) Deletion of Rad16 stabilizes Hrq1 following cisplatin exposure. Hrq1-9xMYC expressing wild-type (WT) or *rad16Δ*, cells were incubated with cycloheximide in the presence or absence of 100 μg/ml cisplatin and/or 0.1% DMSO. Note blot from **Fig 1C** was reshown for comparison. (B) Quantification of the proportion of Hrq1 remaining relative to time 0 (before CHX addition) and the loading control, Kar2, are plotted on the graph in log scale from WT and *rad16Δ* cells. Each experiment was performed three to 5 times with standard error plotted (Raw densitometry data in Sheets F-H in **S1 Data**). Note that the WT cisplatin treated time course is replotted from **Fig 1C**, for direct comparison to *rad16Δ* cisplatin treated cells.

from equal cell numbers every 20 minutes and Hrq1 protein levels analyzed and compared to Kar2 as a loading control and Clb2 as a G2/M regulated cyclin. Consistent with a role during replication, Hrq1 protein levels begin to increase in S/G2 at 40 min after alpha factor release and then plateaus whereas Clb2 protein peaks starting at 60 min (**Fig 3A**). Cell cycle progression was confirmed by FACS analysis (**Fig 3A**). As Hrq1's expression is increased during S/G2, this suggests that it may have a role in during replication where PRR also functions.

Previous studies in plants, fission yeast, and budding yeast have confounding results as to whether Hrq1 functions during PRR [44–46]. Therefore, it remains controversial as to the role of Hrq1 during crosslink repair. To determine if Hrq1 has a role in PRR, we carefully and systematically examined the genetic relationship between *HRQ1* and multiple members of the PRR pathway. We combined *HRQ1* mutants with disruption of PRR genes and tested these mutants for increased cisplatin sensitivity. Since Rad6/Rad18 ubiquitylates Pol30 during early PRR steps (**Fig 3B**), we began our analysis with *rad6Δ* cells. When compared to wild-type, *rad6Δ* cells are sensitive to 2.5 μg/ml cisplatin whereas *hrq1Δ* cells are sensitivity to 30 μg/ml cisplatin. Yeast disrupted with both *HRQ1* and *RAD6*, exhibit sensitivity comparable to *rad6Δ* alone (**Fig 3C**). These results suggest that *HRQ1* likely functions in the PRR pathway downstream of *RAD6*.

After Rad6/Rad18 mono-ubiquitinates PCNA, PCNA becomes poly-ubiquitinated on lysine 164 during error-free PRR (**Fig 3B**). At the same time, PCNA can be independently

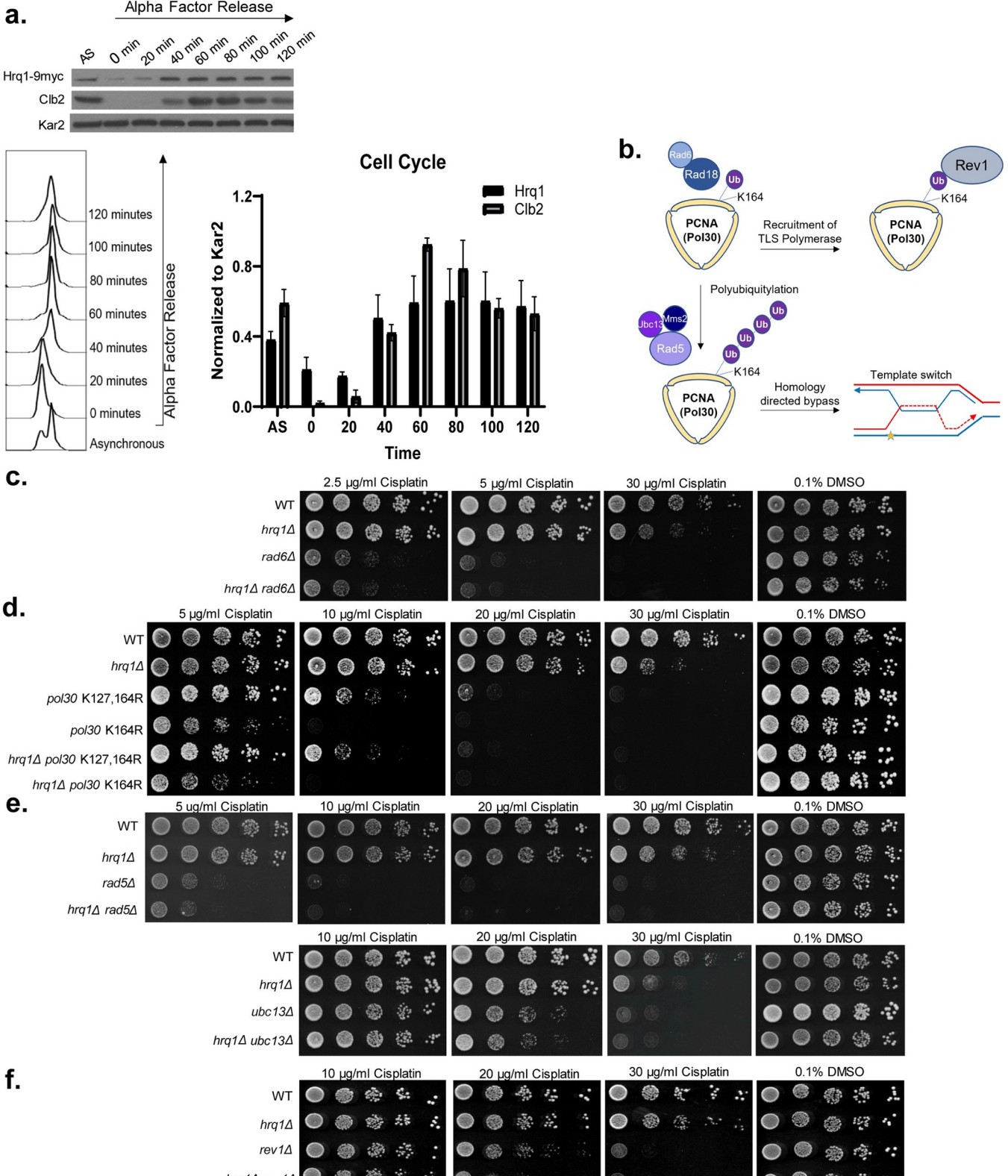

**Fig 3. Hrq1 functions during error-free post-replicative repair.** (A) Hrq1 expression increases during S/G2 and then plateaus. Hrq1-9xMYC expressing cells were either untreated (asynchronous, AS) or cell cycle arrested in G1 with α-factor. The α-factor arrested cells were subsequently released into fresh YPD

medium (0 min) and grown for 120 min. Protein samples from the indicated time points were analyzed by western blot for Hrq1 (anti-MYC), the G2/M cyclin, Clb2, (anti-Clb2), or a loading control, Kar2 (anti-Kar2). Quantification from three experiments, the mean with SEM is shown (Raw densitometry data in S2 Data). The cell cycle stage was analyzed FACS. (B) Schematic of post-replicative repair pathways. When the replication fork stalls, Pol30 (yellow trimer) is initially monoubiquitylated (Ub, purple circle) via Rad6-Rad18 (light and dark blue circles) on lysine 164 (K164). Monoubiquitylation of PCNA recruits the error-prone translesion synthesis polymerases, i.e. Rev1 (gray circle) to bypass the lesion. Alternatively, PCNA is further poly-ubiquitylated by Rad5-Ubc13-Mms2 (light, medium, and dark purple complex). Polyubiquitylation of PCNA mediates error-free repair through template switching, which is a homolog directed process. A bypass intermediate is shown with the newly synthesized DNA in red and the lesion as a yellow star. (C, D) Hrq1 functions in the same pathway as Rad6 and Pol30. The indicated yeast strains were five-fold serial diluted onto SC medium containing DMSO and/or SC medium containing the indicated amount of cisplatin. The plates were photographed after 2 days of incubation at 30°C in the dark. (E) Hrq1 functions in the same pathway as Rad5 and Ubc13. Serial dilutions of the indicated yeast strains were performed as described in (C). (F) Hrq1 functions in a different pathway than Rev1. Serial dilutions of the indicated yeast strains were performed as described in (C).

sumoylated on lysine 127, which recruits the DNA helicase Srs2 (not drawn). To address whether *HRQ1* functions downstream of *PCNA* ubiquitylation, we examined the genetic relationship between *HRQ1* and *PCNA* ubiquitylation/sumo mutants, *POL30-K164R* and *POL30-K127R/K164R*. Disruption of *HRQ1* in combination with *POL30-K164R or POL30-K127R/K164R*, results in cisplatin sensitivity comparable to either *POL30* single mutants (Fig 3D). These results are consistent with *HRQ1* functioning downstream of *POL30* ubiquitylation/sumoylation during PRR.

We next asked whether Hrq1 functions in the "error-free" template switching or "error-prone" translesion synthesis branches of PRR. To do this, we first examined the genetic relationship between *HRQ1* and members of the "error-free" branch of PRR, *RAD5-UBC13-MMS2*, which polyubiquitylates PCNA (Fig 3B). Cells with both *HRQ1* and *RAD5* disrupted exhibit cisplatin sensitivity comparable to a *RAD5* single mutant (Fig 3E). These findings are consistent with a plant study demonstrating that *RAD5* functions in the same pathway as *HRQ1*, and contrast with a budding yeast study, where *HRQ1* was found to function independently of *RAD5* [44–46]. Suggesting that *HRQ1* does indeed function during TS, we observe similar genetic results with another PRR member, *UBC13*, where *hrq1Δ ubc13Δ* double mutants exhibit cisplatin sensitivity to *ubc13Δ* (Fig 3E). Together, these results suggest that *HRQ1* functions in the "error-free" branch of PRR.

Next, we determined if *HRQ1* also functions in the "error-prone" translesion synthesis branch of PRR by examining the genetic relationship between *HRQ1* and *REV1*, a translesion synthesis polymerase. We observe that *hrq1Δ rev1Δ* double mutant cells exhibit increased cisplatin sensitivity in comparison to either a *hrq1Δ* or a *rev1Δ* single mutant (Fig 3F; 20 µg/ml cisplatin). These results suggest that Hrq1 functions primarily in the "error-free" branch of PRR.

## Hrq1 function and regulation in ICL repair is distinct from intrastrand crosslink repair

Loss of *HRQ1* results in sensitivity to different types of DNA crosslinking agents including cisplatin and MMC. While both cisplatin and MMC cause ICLs and intrastrand crosslinks, cisplatin damage results in 90–95% intrastrand crosslinks whereas mitomycin C results in 90–95% ICLs. Therefore, by using MMC, we asked if Hrq1 has a similar function during ICL repair when ICLs are the predominant lesion. To address this, we examined *hrq1Δ* cells for sensitivity to MMC either alone or in combination with a *rev1Δ* or *ubc13Δ* mutant (S3 Fig). As reported in a prior study [21], *hrq1Δ* cells are MMC sensitive (S3A Fig). Like cisplatin induced DNA damage, the *hrq1Δ rev1Δ* double mutant exhibits increased MMC sensitive compared to the single mutants (S3A Fig). These results suggest that *HRQ1* and *REV1* are functioning in different pathways upon MMC exposure. However, in contrast to what we observed with cisplatin damage, a *hrq1Δ ubc13Δ* double mutant exhibits increased MMC sensitivity compared

to the single *hrq1Δ* or *ubc13Δ* mutants (S3A Fig). These results suggest that Hrq1 function in ICL repair may be functionally distinct from intrastrand crosslink repair.

Since we observe genetic differences in *HRQ1* response to cisplatin- and MMC-induced DNA damage, we asked whether Hrq1 protein levels are similarly regulated upon MMC exposure by performing cycloheximide chase experiments (S3B Fig). Quite surprisingly, and in contrast to the cisplatin-induced response, Hrq1 protein levels are largely stable following MMC exposure (compare S3B Fig to Fig 1C). While it is possible that higher doses of MMC may result in Hrq1 degradation, together our genetic and molecular results suggest that Hrq1 function and regulation in response to ICLs is indeed distinct from intrastrand crosslinks.

To validate Hrq1 role in mediating repair of intrastrand crosslinks during PRR, we performed genetic analysis using UV-C, which induces primarily 6–4 photoproducts and cyclobutane pyrimidine dimers. Suggesting a role in template switch, we find that a *hrq1Δ rad5Δ* double mutant has similar sensitivity to UV-C compared to *rad5Δ*. We observe similar results with *UBC13*, as a *hrq1Δ ubc13Δ* double mutant has similar sensitivity to UV-C as a *ubc13Δ* single mutant (S3C Fig). Suggesting *HRQ1* functions in a different pathway to TLS, a *hrq1Δ rev1Δ* double mutant exhibits increased UV-C sensitive compared to the single mutants (S3D Fig). Together these results strengthen the conclusion that Hrq1 functions during intrastrand crosslink repair via PRR.

## Preventing Hrq1 degradation results in increased recombination and mutations

Since we find that Hrq1 functions during "error-free" PRR during intrastrand crosslink repair and is regulated by the 26S proteasome, we sought to determine if misregulation of Hrq1 protein levels alter PRR. To do this, we determined the lysine residues that catalyze Hrq1 degradation upon cisplatin exposure (Fig 4A). Hrq1 is 1077 amino acids and contains 77 lysines. To identify potential ubiquitylation sites, we analyzed Hrq1 protein using UbPred, which predicts five ubiquitylated lysines (K164, K219, K221, K366, K872). In addition, we included two lysines that are conserved between Hrq1 and its mammalian ortholog, RECQL4 (K839, K938), as well as one lysine (K366), which is both predicted to be ubiquitylated by UbPred and conserved with RECQL4 [54,55]. We mutated these seven lysine residues to arginine at the endogenous *HRQ1* locus and herein refer to this mutant as Hrq1-7KR. We next determined whether mutating these Hrq1 lysine residues results in Hrq1 protein stabilization. To test this, we performed cycloheximide chases on Hrq1-3xHA and Hrq1-7KR-6xHA. As previously observed, Hrq1 protein levels are reduced upon cisplatin treatment (**Figs 1C and 4B**). In contrast, Hrq1-7KR protein expression remains similar in both cisplatin treated and untreated conditions (**Fig 4B**). It is interesting to note that the Hrq1-7KR mutant is not fully stabilized, suggesting that additional lysine residues may contribute to Hrq1 degradation independently of its DNA damage response function. These results suggest that the Hrq1-7KR mutant protein levels are misregulated in response to cisplatin.

We have thus far identified a function for Hrq1 in error-free PRR during intrastrand crosslink repair and found that Hrq1 targeted for degradation upon cisplatin treatment. Therefore, we hypothesized that Hrq1 may need to be degraded for completion of PRR. Since Hrq1 functions during template switching, which is a recombination-based pathway, we asked whether stabilization of Hrq1 protein levels leads to increased recombination. To test this, we utilized the Hrq1-7KR mutant strain. At the same time, we created a strain where we induce Hrq1 over-expression (Hrq1 OE) by replacing *HRQ1*'s endogenous promoter with a galactose-inducible/dextrose-repressible *GAL1* promoter. GAL-3xHA-Hrq1 results in an approximately five-fold increase in Hrq1 expression compared to Hrq1-3xHA (S4A Fig). Therefore, we used

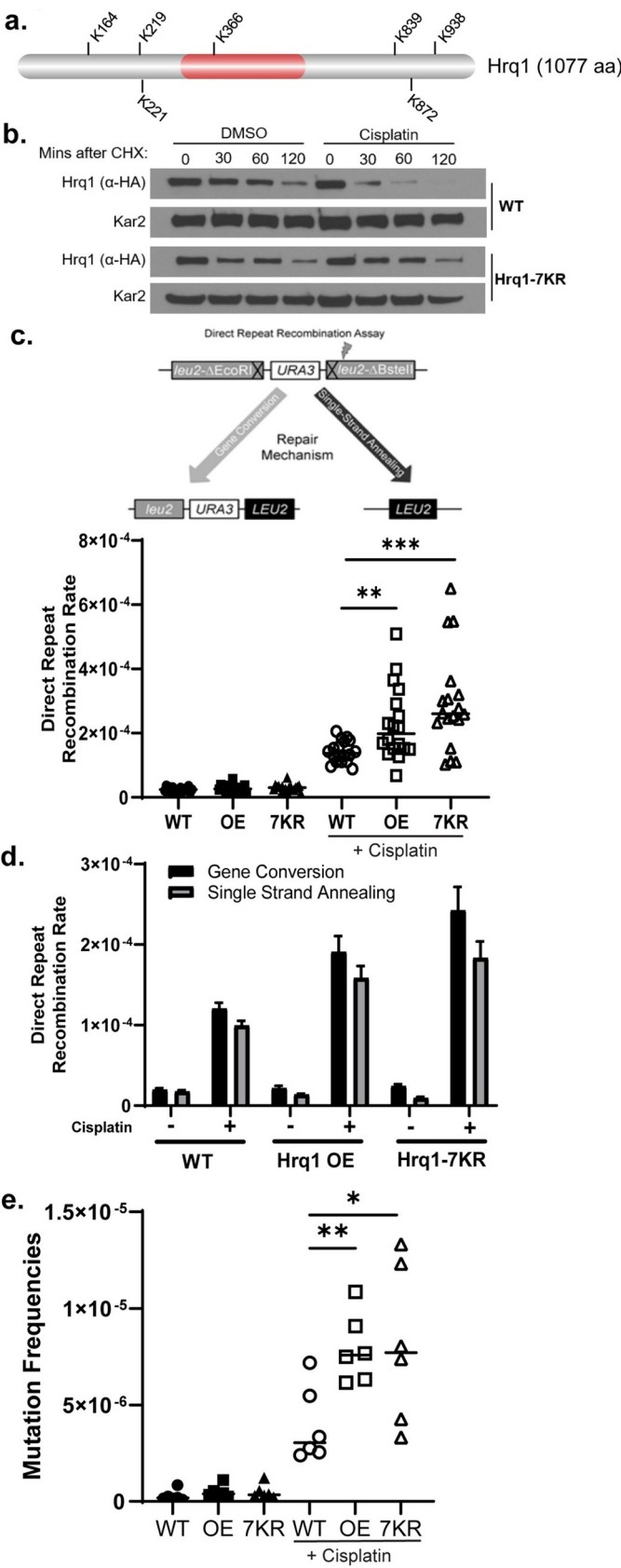

**Fig 4. Hrq1 protein levels are stabilized following cisplatin exposure by mutating the predicted Hrq1 ubiquitylated lysine residues to arginine.** (A) Schematic of Hrq1 with lysines residues that are predicted to be ubiquitylated (K164, K219, K221, K872) or conserved between Hrq1 and RecQL4 (K366, K839, K938) is shown. The helicase domain is indicated in red (287–496 aa, InterPro). (B) Hrq1 protein levels are stabilized in Hrq1-7KR mutant. Cycloheximide chase experiments were performed in Hrq1-3xHA or Hrq1-7KR-6xHA expressing cells. Equal number of cells were collected every 30 minutes for 120 minutes. The experiment was performed in duplicate and a representative image is shown. (C) Overexpression or stabilization of Hrq1 leads to increased recombination following cisplatin treatment (35 μg/ml, 16 hours). Schematic of direct repeat recombination (DRR) assay, where recombinants are measured by formation of *LEU+* recombinants (graphed). In this assay, two *leu2* heteroalleles are disrupted by insertion of an EcoR1 or BsteII restriction sites, respectively, with an intervening *URA3* gene. Restoration of *LEU+* can occur by Rad51-dependent gene conversion (GC) measured by *URA3+ LEU2+* recombinants whereas Rad51-independent single-strand annealing (SSA) is measured by *URA3- LEU2+* recombinants. Nine independent colonies were measured for each experiment and the median value from two experiments (horizontal bar) were plotted (Colony count in **S3** and **S4 Data**). ** represents p<0.01, *** represents p<0.001 (D) The average rates of GC and SSA for each condition are shown, average rates with SEM are graphed. (E) Overexpression or stabilization of Hrq1 leads to increased mutagenesis following cisplatin treatment (30 μg/ml, 16 hours). Each measurement (dots) and the median value from three experiments (horizontal bar) were plotted (Colony count in **S5 Data**). * represents p<0.05, ** represents p<0.01.

a Hrq1-7KR strain characterized above for our studies but treated the cells in the same manner as the GAL-3xHA-Hrq1 cells to enable direct comparisons between the two strains.

Using the Hrq1-7KR-6xHA mutant and the GAL-3xHA-Hrq1 strains, we performed a direct repeat recombination assay in the presence of galactose containing rich medium. We find that over-expression or stabilization of Hrq1 results in a 1.5 to 2-fold increase in total recombination compared to WT cells following cisplatin treatment (**Fig 4C**). The recombinant colonies in the direct repeat recombination assay can be formed by gene conversion or single-strand annealing. We then asked whether increased Hrq1 or Hrq1-7KR results in more gene conversion or single-strand annealing. We observe that both gene conversion and single-strand annealing are increased upon cisplatin exposure in the GAL-Hrq1 and Hrq1-7KR cells compared to wild-type (**Fig 4D**). These results suggest high levels of Hrq1 following cisplatin exposure results in increased homology-directed repair.

Although recombination during template switching is considered "error-free", aberrant template switching can result in increased DNA mutations [56,57]. Therefore, we examined whether Hrq1 over-expression or stabilization increases mutagenesis. To measure mutation rates, we performed a canavanine mutagenesis assay. This assay measures mutations in the *CAN1* permease gene, which enables cell viability upon exposure to the toxic arginine analog, canavanine. We observe over two-fold increase in mutation rates in Hrq1 over-expressing or 7KR cells in the presence cisplatin in comparison to WT cells (**Fig 4E**). These results suggest that Hrq1 must be tightly regulated to prevent excess recombination and mutagenesis.

Lastly, it is possible that preventing the degradation of Hrq1 results in accumulation of toxic DNA repair intermediates. To test this hypothesis, we examined the sensitivity of Hrq1-7KR cells upon cisplatin exposure (**S4B Fig**). Surprisingly, we do not observe decreased cell viability of Hrq1-7KR cells compared to WT (**S4B Fig**). This suggests that stabilization of Hrq1 during PRR may result in DNA repair intermediates that are ultimately resolved using an alternative pathway. Although we do not observe cisplatin sensitivity in the Hrq1-KR mutant, overexpression of the HRQ1-7KR (by using a galactose inducible promoter) results in cell lethality even in the absence of DNA damage (**S4C Fig**).

## Similar to Hrq1, the levels of RECQL4 decreases following treatment to DNA crosslinking agents

We next sought to determine whether there is a conserved regulatory role for human RECQL4 following cisplatin exposure. We used U2OS cells since they are commonly used in DNA

repair studies and Rothmund-Thomson patients with *RECQL4* mutations are predisposed to osteosarcomas. Since RECQL4 is needed for cisplatin resistance, we hypothesized that similar to Hrq1 its protein levels may also decrease follow cisplatin exposure. To test this hypothesis, we repeated our cycloheximide chase experiments in the U2OS cell line. Similar to yeast Hrq1, RECQL4 protein levels decrease following cisplatin exposure in comparison to the untreated control (**Fig 5A**).

We next determined whether this phenomenon holds true with an endogenous crosslinking agent, acetaldehyde. Acetaldehyde is known as the toxic byproduct of alcohol metabolism, but it is also naturally produced from the breakdown of various foods including yogurt, apples, etc. [58,59]. Acetaldehyde is highly reactive and can produce similar lesions to cisplatin [60,61]. Like cisplatin, RECQL4 levels decrease following acetaldehyde exposure (**Fig 5B**). Suggesting that RECQL4 degradation is also lesion specific, a prior study showed that RECQL4 protein levels are maintained following IR or camptothecin exposure [62]. Together, these results suggest that there is a conserved regulatory mechanism between yeast Hrq1 and human RECQL4 during DNA crosslink repair.

Since we find that Hrq1 functions in the error-free branch of PRR, we hypothesized that RECQL4 functions similarly. To test this, we determined the cisplatin sensitivity of U2OS cells with siRNA knockdown of *RECQL4* alone or in combination with the error-prone PRR polymerase, *REV1*. Using an MTS assay, we find that both *siRECQL4* and *siREV1* individually treated cells are cisplatin sensitive (**Fig 5C**). If *RECQL4* and *REV1* function in the same PRR branch, we expect that siRNA knockdown of both *RECQL4* and *REV1* will result in similar cisplatin sensitivity as the single knockdowns. However, if *RECQL4* and *REV1* function in different pathways, we expect increased cisplatin sensitivity when both *RECQL4* and *REV1* are knocked down. Consistent with *RECQL4* and *REV1* functioning in different pathways, we observe increased cisplatin sensitivity in the double treated *siRECQL4 siREV1* cells using both MTS and clonogenic survival assays (**Fig 5C and 5D,** respectively). These results suggest that, like *HRQ1*, *RECQL4* may function in the error-free PRR pathway. In the future, the genetic relationship between *RECQL4* and genes of the TS pathway should be assessed to determine whether RECQL4 truly has a function during TS.

Since yeast Hrq1 over-expression led to increased homology-directed repair, we examined whether over-expression of human RECQL4 also leads to increased recruitment of the recombinase, RAD51, to DNA repair foci. To do this, we created a plasmid that over-expressed RECQL4 (**S4D Fig**) and measured these cells for RAD51 focus formation in the presence or absence of cisplatin. RAD51 is a central homologous recombination protein that is relocated to DNA repair sites following DNA damage. Suggesting that there is increased recombination, we find that RECQL4 over-expression results in a significant increase in the number of cells with RAD51 foci in both untreated and cisplatin treated cells (**Fig 6A;** p < 0.0001 for both). Since RECQL4 is a DNA helicase, we asked whether its helicase activity would be crucial for RAD51 focus formation upon cisplatin exposure. Suggesting that RECQL4 DNA unwinding activity likely contributes to RAD51 focus formation, we observe that that RECQL4 helicase mutant, RECQL4-K508A, exhibits reduced RAD51 foci compared to RECQL4 over-expressing cells (**Fig 6A**; p <0.01). The increased RAD51 foci observed could be due to hyper-recombination as well as increased DNA damage or persistent RAD51 foci formed from unresolved recombination intermediates. Therefore, the RECQL4 protein family has a conserved function in recombination during DNA crosslink repair.

The RAD51 foci observed upon RECQL4 over-expression could be indicative of increased DSB formation, recombination, or even ssDNA from excess RECQL4-mediated DNA unwinding. To examine whether RECQL4 overexpression leads to increased DSB formation, we performed neutral comet assays. We find that RECQL4 over-expression in untreated or

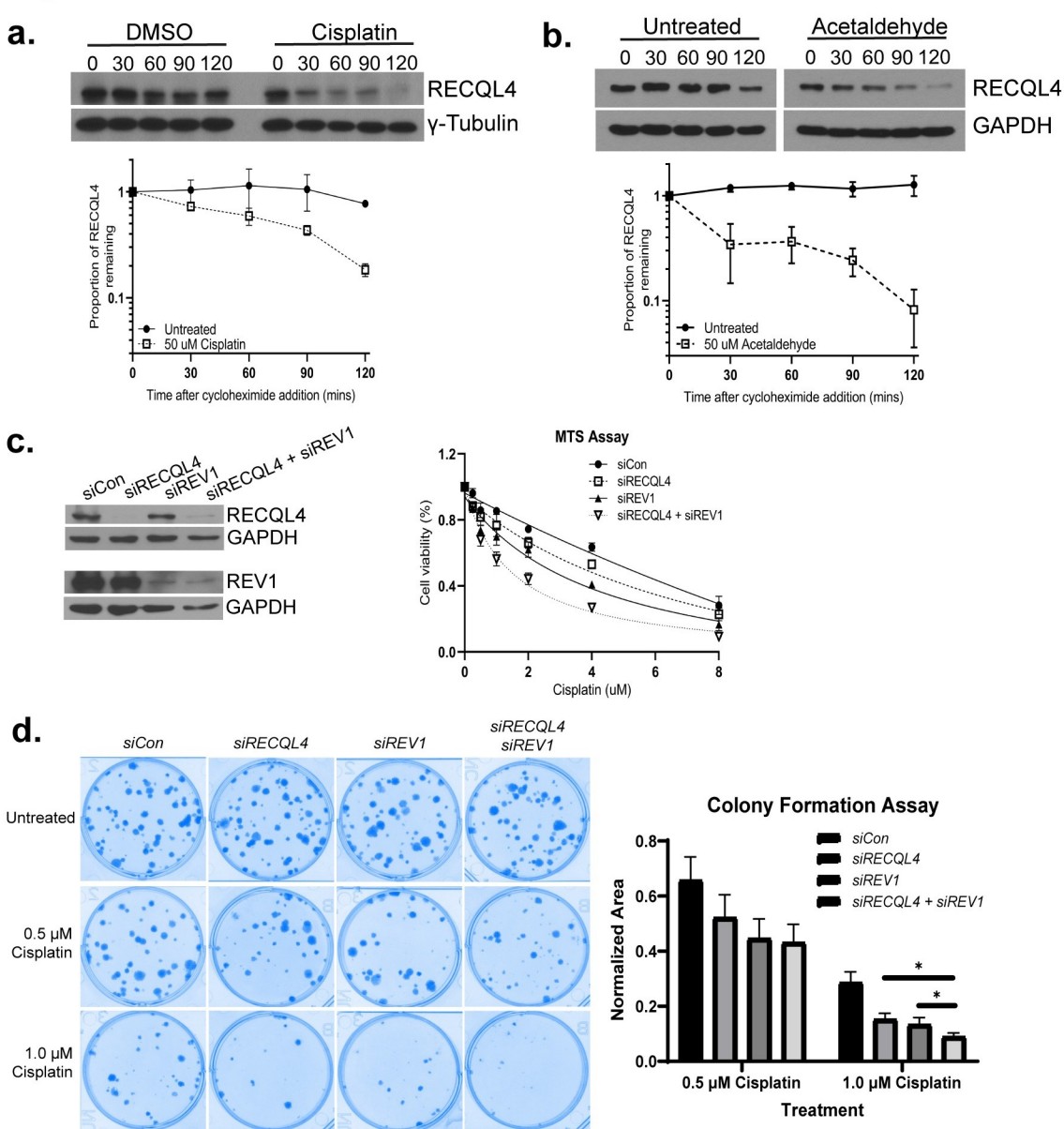

**Fig 5. RECQL4 functions in a separate pathway from REV1 and is degraded following DNA crosslinks.** (A) RECQL4 levels decrease following cisplatin treatment. U2OS cells were incubated with 50 μg/ml cycloheximide in the presence or absence of 50 μg/ml cisplatin. Protein extracts were analyzed by western blot for endogenous RECQL4 protein levels (α-RECQL4) or a loading control, Tubulin (γ-Tubulin), at the indicated time points. Quantification of the proportion of RECQL4 remaining relative to time 0 (before CHX addition) and the loading control, Tubulin, are plotted on the graph in log scale. Each experiment was performed in triplicate with standard error plotted (Raw densitometry data in Sheets A-C in **S6 Data**). (B) RECQL4 protein levels decrease following acetaldehyde treatment. U2OS cells were incubated with 50 μg/ml cycloheximide in the presence or absence of 50 μg/ml acetaldehyde. Protein extracts were analyzed by western blot for endogenous RECQL4 protein levels (α-RECQL4) or a loading control, GAPDH (α-GAPDH), at the indicated time points. Quantification of the proportion of RECQL4 remaining relative to time 0 (before CHX addition) and the loading control, GAPDH, are plotted on the graph in log scale. Each experiment was performed in triplicate with standard error plotted (Raw densitometry data in Sheets D-F in **S6 Data**). (C) Cisplatin exposed U2OS cells exhibit decreased viability when both RECQL4 and REV1 are knocked down by siRNA. MTS assay was performed on after 72 hr exposure to the indicated dose of cisplatin. Western is shown to confirmed knockdown of proteins, note since RECQL4 and REV1 are of similar size, the same samples were loaded twice to detect RECQL4 and REV1. Each experiment was performed three times with mean and standard errors plotted (Unprocessed data in **S7 Data**). (D) Decreased cell viability following cisplatin exposure when both RECQL4 and REV1 are knocked down. Clonogenic survival assay with siControl (scrambled siCon), siRECQL4, siREV1, and siRECQL4 siREV1 treated cells with or without cisplatin. Each condition was plated in triplicate for each experiment and the experiment was performed three times with mean and standard errors plotted (Colony quantification in **S8 Data**).

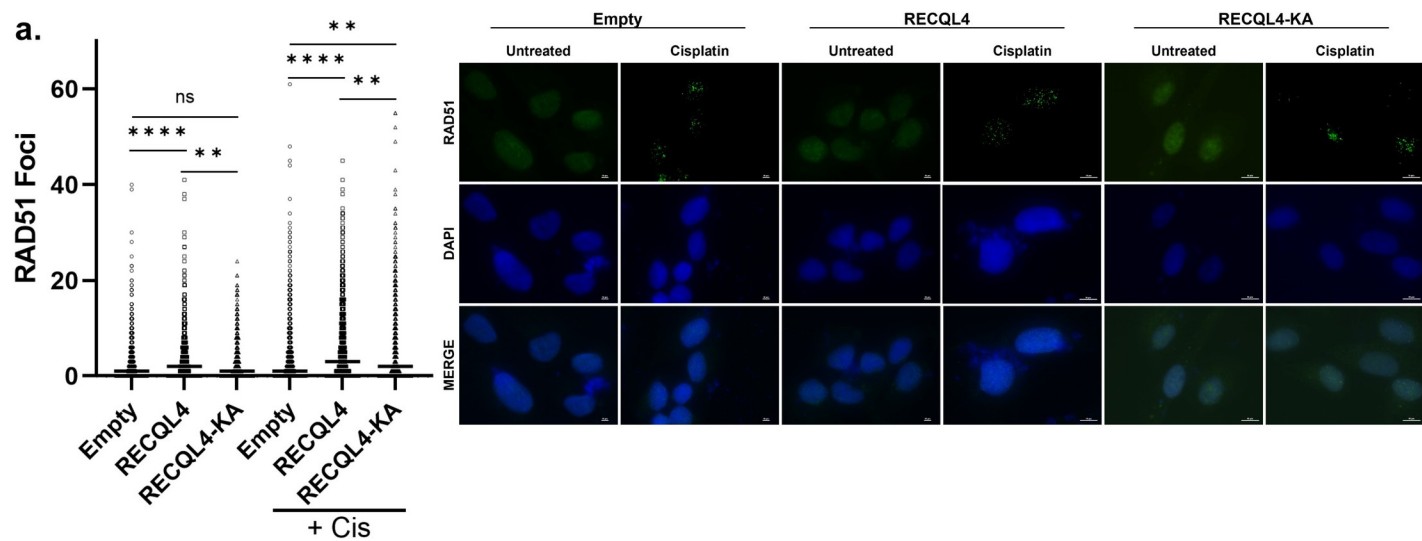

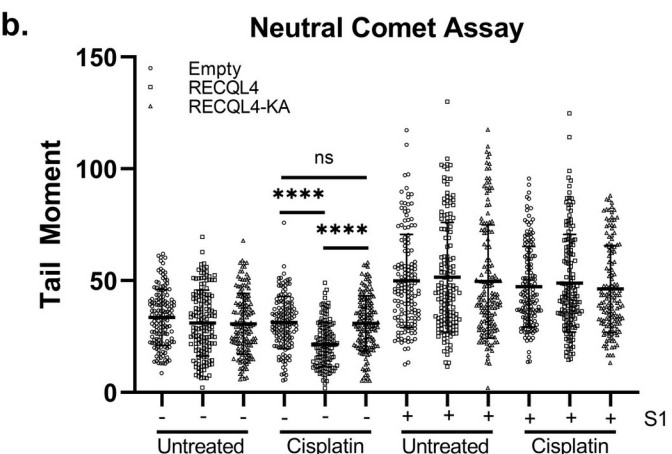

**Fig 6. Overexpression of *RECQL4* results in increased RAD51 foci and decreased tail moment.** (A) Overexpression of *RECQL4* results in increased RAD51 foci, which is dependent on its helicase activity. U2OS cells were transfected with an empty plasmid or a plasmid expressing RECQL4 or RECQL4-K508A under a CMV promoter. The cells were either mock or cisplatin treated for one hour and after a two-hour recovery, imaged for RAD51 foci or DAPI by immunofluorescence. RAD51 foci was quantified from 200 cells per condition for each experiment. The experiment was performed three to five times and the median was graphed (Unprocessed foci count in **S9 Data**). Representative images are shown. (B) Overexpression of *RECQL4* results decreased tail moment following cisplatin exposure, which is dependent on its helicase activity. U2OS cells were treated similarly to the immunofluorescence experiment, before being harvested for neutral comet assay. At least 40 comets were counted per condition for each experiment. The experiment was performed four times and the mean and standard deviation was graphed (Unprocessed tail moments in **S10 Data**).

cisplatin-treated cells do not increase tail moments compared to the empty control U2OS cell line (**Fig 6B**). Intriguingly, RECQL4 over-expression decreases tail moments upon cisplatin exposure.

Suggesting that RECQL4 helicase activity is critical to prevent DSB formation following cisplatin, the RECQL4 helicase mutant, RECQL4-K508A, has similar tail moments to the empty control (**Fig 6B**). These finding are consistent with a role for RECQL4 in preventing DSB formation and fork collapse, perhaps, by enabling lesion bypass through TS.

It is possible is that the increased RAD51 foci observed may be due to more ssDNA caused by RECQL4 DNA unwinding. To examine whether RECQL4 over-expression results in more ssDNA, we quantitated the tail moments following S1 nuclease treatment. If more ssDNA is present, the S1 nuclease will degrade the ssDNA resulting in more DSBs. Although statistically insignificant, we find that RECQL4 over-expression extremely modestly increases ssDNA and

that this increase is helicase dependent (**Fig 6B**). Note that since we are assessing RECQL4 role during replication, and not DSB repair, we did not treat the cells long enough with cisplatin to induce DSBs. In this treatment timeframe, the RAD51 foci that form are likely due to RAD51 function during replication. Overall, our studies suggest that RECQL4 over-expression does not significantly increase either DSBs or ssDNA. A possible technical weakness of the S1 neutral comet assay is that long stretches of ssDNA could be coated with the ssDNA binding protein, replication protein A (RPA), which could hinder S1 digestion. Thus, it is still possible that RECQL4 over-expression may lead to increased ssDNA and future studies can address this looking at RPA foci. However, since we did not observe an increased amount of DSBs, the increased RAD51 foci observed upon RECQL4-over expression could be due to increased recombination happening at the replication fork.

## Bioinformatic analysis reveals elevated RECQL4 levels are associated with tumorigenesis and increased tumor mutation burden

We find that Hrq1 over-expression results in genomic instability, which is a cancer hallmark. Therefore, we asked whether RECQL4 is over-expressed in breast tumors relative to normal tissue. By examining mRNA levels of *RECQL4* in TCGA, we observe that *RECQL4* expression is increased in tumors versus normal tissues (**Fig 7A**). Several tumor types where RECQL4 is over-expressed are treated with cisplatin, including breast cancer. Therefore, we focused on breast cancer where we can delineate between breast cancers that are classically treated with cisplatin versus those that are not. Estrogen receptor negative (ER-) breast cancers are generally more aggressive and typically treated with cisplatin including TNBC, an ER- breast cancer subtype [63]. In contrast, ER+ breast cancers are classically treated with hormone therapy [63]. When compared to ER+ breast cancers, ER- breast cancers have elevated levels of *RECQL4* expression (**Fig 7B**). Our result is consistent with another study where *RECQL4* expression was elevated in more aggressive cancers [14].

As elevated Hrq1 protein levels in yeast result in increased mutations, we postulated that increased human *RECQL4* expression correlates with increased tumor mutation burden. Indeed, when examining ER- cancers, high levels of *RECQL4* correlates with increased mutational burden (**Fig 7C**).

When yeast Hrq1 is over-expressed, we find that both mutations and recombination are increased. Therefore, we asked whether over-expression of RECQL4 results in increased expression of recombination genes. Using the METABRIC and TCGA mRNA expression datasets, we examined which cellular pathways are enriched in RECQL4 over-expressing ER- tumors. This analysis revealed that cell cycle progression, transcriptional regulation, and DNA repair are enriched in RECQL4 high subset of TNBC (**Fig 7D**). Upon further analysis, we find that high levels of *RECQL4* significantly correlates with genes in the HR pathway and RAD51 (**Fig 7E**). These results are consistent with our yeast and mammalian studies here suggesting that high levels of *RECQL4* correlate with increased mutations and perhaps increased recombination.

In tumor cells, DNA repair defects can result in therapeutic sensitivity. Since *RECQL4* over-expression leads to increased genomic instability, we asked whether *RECQL4* expression may predict therapeutic response to cisplatin. To do this, we analyze a study where they examined cisplatin-treated TNBCs, tumors with elevated *RECQL4* expression have a better clinical response compared to tumors with lower *RECQL4* expression (**Fig 7F and 7G**) [64]. This suggests that increased *RECQL4* levels may provide a prognostic marker for therapeutic response to cisplatin. It should be noted that the publication [64] used a small sample size (n = 24), so more studies are needed to validate these results.

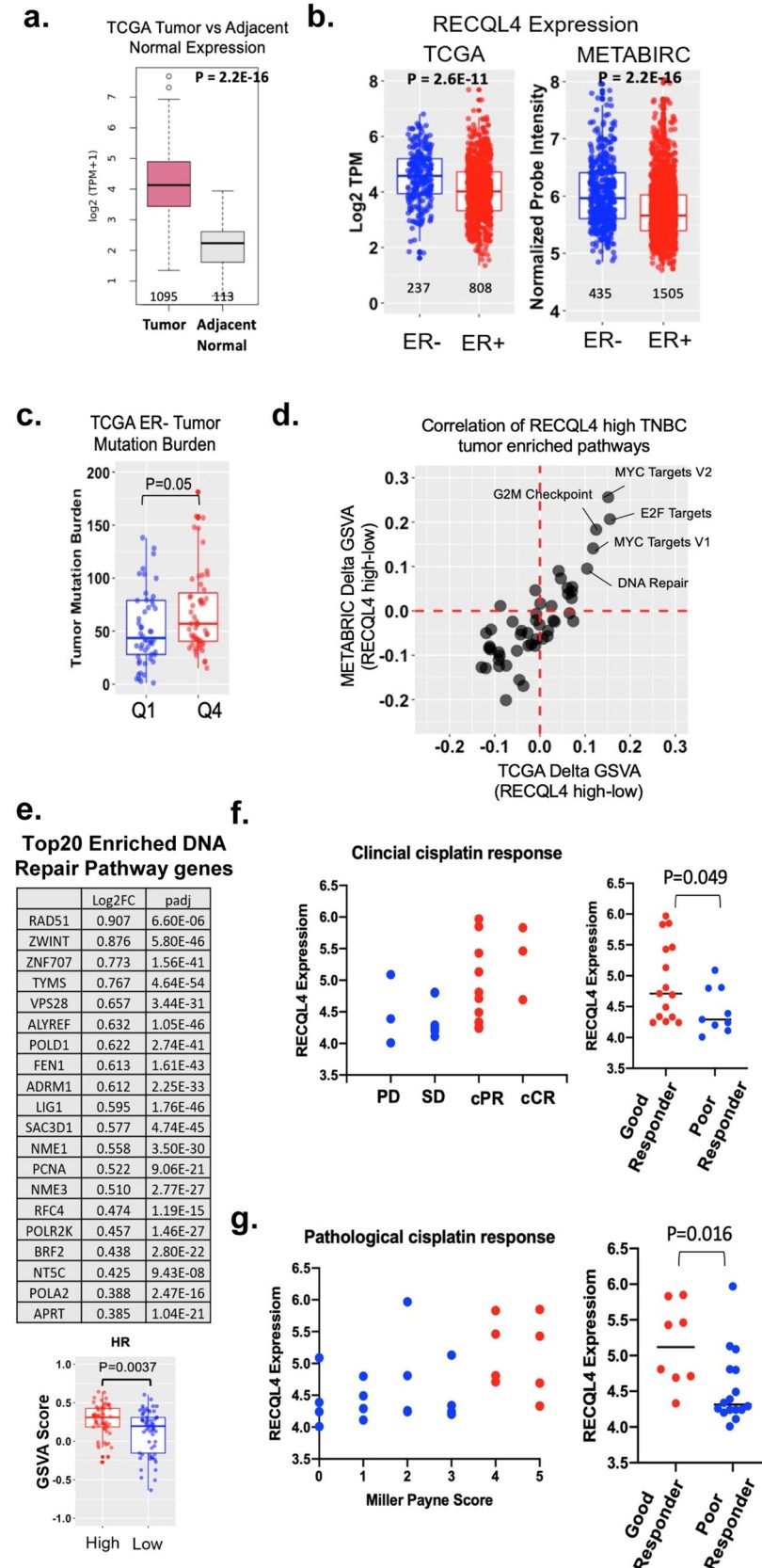

**Fig 7. High RECQL4 expression correlates with increased recombination, mutations, and tumorigenesis.** (A) RECQL4 expression is elevated in tumors in comparison to adjacent normal tissue. RNAseq data from tumors and normal tissue matched samples was acquired from TCGA. The data was normalized by transcripts per million (TPM), then analyzed for *RECQL4* expression. Box and whisker plots graph the median of the data alongside the 25th and 75th percentile. The number of samples alongside p-value are shown using Mann-Whitney U test. (B) RECQL4 expression is higher in ER- breast cancer. Transcriptome data from TCGA (RNA-seq) and METABRIC (microarray) was analyzed for *RECQL4* expression in ER- (blue) and ER+ (red) cancers. Log2 (TPM+1) and normalized probe intensity were used respectively. Box and whisker plots alongside each data point was graphed. The number of samples alongside p-value are shown using Mann-Whitney U test. (C) Elevated levels of RECQL4 are associated with increased tumor mutation burden. The expression of RECQL4 in ER- breast tumors from TCGA was divided into quartiles based upon expression level. The mutation burden between the least expressing tumors (Quartile 1 –Q1, blue) and the highest expressing tumors (Quartile 4 –Q4, red) was analyzed. Box and whisker plots alongside each data point was graphed. The number of samples alongside p-value are shown. Annotated mutational data from ER- breast cancer was acquired from Firebrower. The mutational data was categorized using "maftools" in R. Tumor mutation burden was scored as described (Wang *et al* 2019). Significance was determined by Mann-Whitney U test. (D) High expression of *RECQL4* correlates with enrichment of DNA repair, G2M checkpoint, MYC Targets V1, E2F Targets 1 and 2 in TNBC. Gene set variation analysis (GSVA) was performed on TNBC gene expression data from both TCGA and METABRIC using the 50 Hallmark gene sets from MSigDB. Correlation between TCGA and METABRIC datasets was plotted for enriched pathways. (E) Homologous recombination and RAD51 is enriched in TNBC tumors expressing high levels of *RECQL4*. GVSA results of "KEGG Homologous Recombination" gene set (MSigDB M11675) was parsed for the top 20 DNA repair genes enriched in TNBC tumors expressing high levels of *RECQL4*. HR pathway was enriched by GSVA analysis in RECQL4 high expressing TNBC compared to loss expressing TBNC. Box and whisker plots alongside each data point was graphed. The number of samples alongside p-value are shown using Mann-Whitney U test. (F,G) *RECQL4* expression serves as a predictor for clinical outcomes. High expression of *RECQL4* correlates with positive response to cisplatin and favorable clinical outcome (Good responder). Analysis from a study where cisplatin was given as a neoadjuvant therapy in 24 TNBC tumors [64]. Lower expressing RECQL4 tumors are shown as blue dots and high expressing RECQL4 tumors are shown as red dots. Clinical response to cisplatin was measured as progressive disease (PD), stable disease (SD), partial response (cPR), complete response (cCR). Good and poor responders were defined as patients with cPR/cCR status and PD/SD status respectively. Increased RECQL4 expression correlates with a good clinical response as determined by Mann-Whitney U test. Pathological response to cisplatin was monitored by Miller Payne metric score (grade 1 is no significant tumor reduction and grade 5 is compete tumor reduction) and graphed based on RECQL4 expression. Good and poor responders were defined as patients with a Miller Payne Score of 4–5 or 0–3 respectively. Increased RECQL4 expression correlates with a good pathological response as determined by Mann-Whitney U test.

## Discussion

DNA crosslinks can arise endogenously from reactive aldehydes, or exogenously from chemotherapeutic agents such as cisplatin. DNA crosslink repair is critical to maintain genomic stability and for disease prevention. While the RECQL4 family has a conserved role in DNA crosslink repair, its role during this process is poorly understood. Here we use the yeast model to identify a function for the RECQL4 family in DNA crosslink repair by placing yeast Hrq1 in the template switching pathway to mediate bypass of intrastrand crosslinks. Furthermore, we find that Hrq1 is regulated by the UPS during intrastrand crosslink repair and that misregulation of Hrq1 protein levels results in increased recombination and mutagenesis. We extend this analysis to mammalian cells and show that like yeast, human RECQL4 protein levels are depleted in response to DNA crosslinks. Using bioinformatic/meta-analysis, we show that increased *RECQL4* levels is associated with increased tumor burden, mutations, and therapeutic response. Together our studies demonstrate a conserved role for the RECQL4 protein family in crosslink repair whose levels must be fine-tuned to prevent excess recombination and mutations.

Our results show that Hrq1/RECQL4 protein levels are tightly regulated to ensure accurate repair of DNA intrastrand crosslinks. Protein degradation plays a crucial regulatory role during DNA repair. While both Hrq1 and RECQL4 are needed for resistance to crosslinking agents, their protein levels decrease upon exposure to cisplatin or acetaldehyde (**Figs 1 and 5**). Consistent with this observation, proteasome-mediated degradation of other DNA helicases (BLM, FBH1, HLTF, amongst others) was observed following DNA damage [65]. Degradation

of these key helicases is critical for dictating which pathway is used as well as preventing accumulation of toxic DNA repair intermediates. Thus, degradation of DNA helicases serves an important role in maintaining genome homeostasis. A prior study suggested that RECQL4 is not targeted by the ubiquitin proteasome in HeLa cells even though it directly interacts with the E3 ubiquitin ligases, UBR1 and UBR2 [66]. However, this study was not performed with DNA damage. Therefore, it is possible that RECQL4 may be ubiquitylated in a DNA damage-dependent manner or that there could be cell type specific differences. Consistent with this notion, recent studies demonstrate that RECQL4 ubiquitylation plays a critical role in double-strand break repair [67,68]. Indeed, a prior study found that RECQL4 ubiquitylation leads to its accumulation at DSB sites, instead of triggering its degradation [67]. However, it is important to note that RECQL4 role in DSBR was being assessed whereas we analyzed its function during replication. Furthermore, RECQL4 protein levels were not assessed in these studies, suggesting that it is still possible that RECQL4 may be degraded in response to DSBs [67,68]. In fact, a recent paper demonstrated that RECQL4 is ubiquitylated and degraded following treatment with bleomycin, which can induce DSBs [69]. However, an earlier study analyzed RECQL4 protein level following IR or camptothecin treatment and found that it was unchanged [62]. The differences between these studies could be due to the different cell types that were used as well as different DSBs inducing agents. Further studies are needed to determine whether RECQL4 is truly degraded following formation of DSBs. It is important to note that in budding yeast, Hrq1 proteasomal degradation is specific for DNA intrastrand crosslinks (**Figs 1B** and S3B).

While our retrospective analysis demonstrated that RECQL4 levels correlate with increased response to cisplatin, it should be noted that multiple studies demonstrated that high levels of RECQL4 results in resistance to cisplatin in other cancer types including breast and gastric cancer cell lines [14,15,64]. It is possible that the increased mutations observed with RECQL4 over-expression elicits a tumor immune response that is not observed in cultured cells or perhaps the small sample size or cohort analyzed may be distinct. Thus, more studies are needed to explain these differences.

Functional analysis of the RECQL4 protein family during crosslink repair is best described for ICLs in yeast. After a DNA intrastrand crosslink occurs during DNA replication, the crosslink can result in fork stalling which leads to mono-ubiquitylation of PCNA by Rad6/Rad18 (**Fig 8**). Subsequently, the lesion is bypassed by the translesion polymerase, Rev1, or alternatively, polyubiquitylation by Rad5-Ubc13-Mms2 promotes a template switching mechanism. Our model shows that Hrq1 mediates template switching and is subsequently ubiquitylated by Rad16 and degraded by the proteasome (**Fig 8**). After the replication fork passes, then NER mediates removal of the intrastrand crosslink. This mechanism is likely conserved in humans as RECQL4 knockdown results in cisplatin sensitivity and altered replication fork dynamics (Arora et al. 2016). Suggesting a conserved function, biochemical analysis demonstrates that both Hrq1 and RECQL4 bind and unwind similar DNA substrates including bubbles, D-loops, and poly(dT) forks [22]. Our studies here demonstrate that like RECQL4, Hrq1 also plays a key role during replication and is cell cycle regulated, begin to increase in S/G2 [70]. Together our results suggest a conserved mechanism for the RECQL4 family during replication-associated intrastrand crosslink repair (**Fig 8**).

Previous studies presented conflicting results as to whether Hrq1 functions during PRR [44–46]. There are multiple explanations for this discrepancy. For example, in *S. pombe* polyubiquitylation of PCNA (Pol30) triggers both TS and TLS. Therefore, the synthetic lethality observed between fission yeast *HRQ1* and factors that mediate PCNA polyubiquitylation may be due to function in TLS rather than an independent function in TS [44]. Separation-of-function alleles are needed to differentiate between these two functions. Our study also contradicts

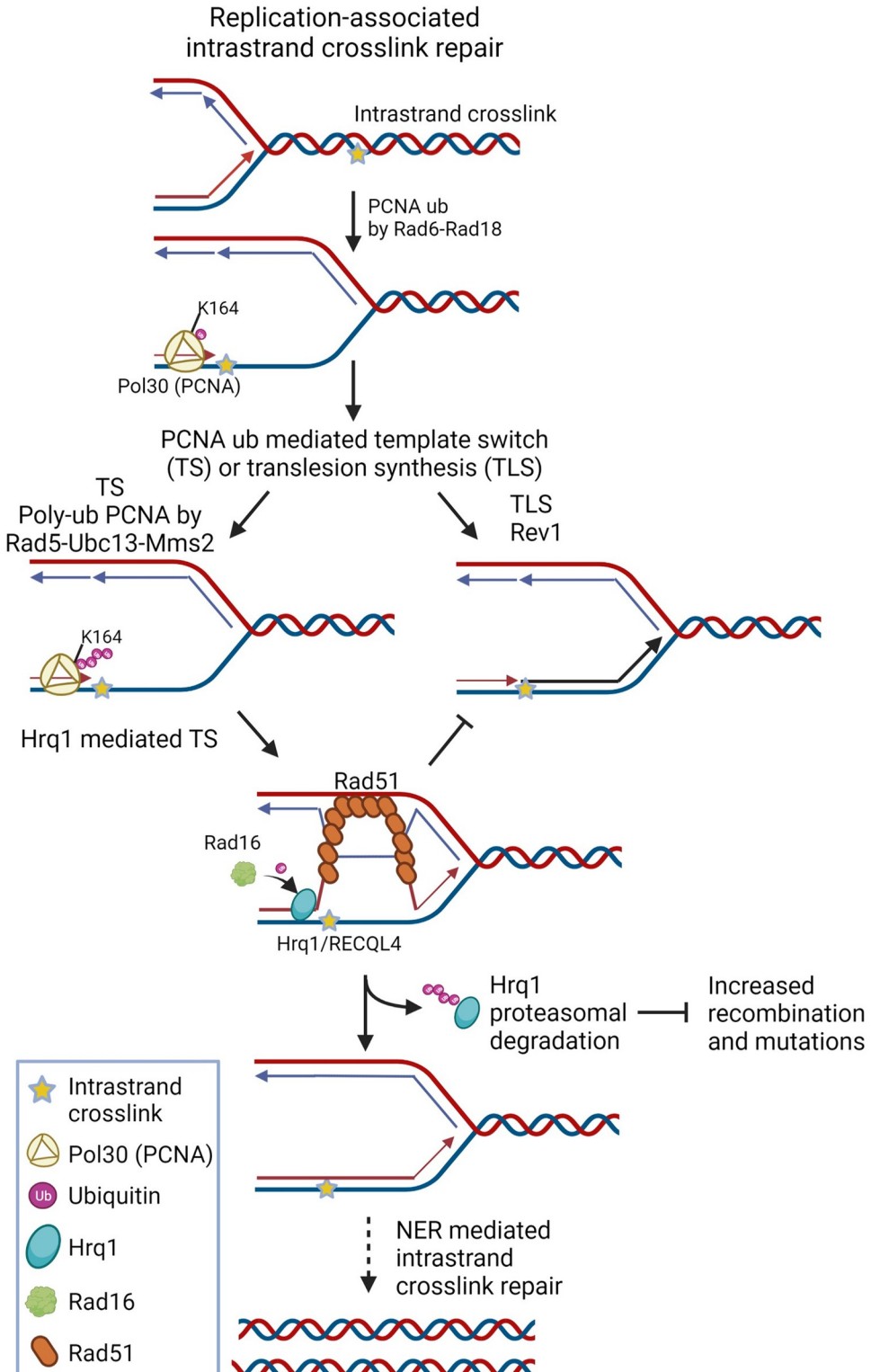

**Fig 8. Model of Hrq1/RECQL4 function during replication-associated intrastrand crosslink repair.** When the replication fork stalls, PCNA (yellow trimer) is monoubiquitylated (Ub, purple circle) by Rad6-Rad18 on lysine 164 (K164). Monoubiquitylation of PCNA recruits the error-prone translesion synthesis polymerases, i.e. Rev1 to bypass the lesion. Alternatively, PCNA is further poly-ubiquitylated by Rad5-Ubc13-Mms2. Polyubiquitylation of PCNA mediates error-free repair through template switching, which is a homology directed process. Hrq1/RECQL4 (blue

oval) facilitates a template switch to bypass the lesion and then is subsequently ubiquitylated by Rad16 and degraded. Degradation of Hrq1 by the proteasome prevents aberrant recombination and subsequent mutation. Once the lesion is bypass, the NER machinery facilitates repair. Created using biorender.com.

another in *S. cerevisiae*, which found that Hrq1 functions independently of PRR [45]. It is possible that there are background strain differences since we used W303 and the other group used the YPH499 genetic background (derived from YNN216). It is important to note that our study here performed extensive genetic analysis at each step of the PRR pathway whereas the other study only analyzed *RAD5* and *REV3* mutants [45]. Furthermore, we find that our results are consistent with UV-C damage as well (**S3C Fig**). Lastly, we note that our results are consistent with those from plants where Hrq1 was found to function in PRR [46] and with our genetic analysis in mammalian cells (**Fig 5**). Altogether, we provide strong evidence that Hrq1 promotes recombination during TS to mediate repair of intrastrand crosslinks.

## Materials and methods

### Yeast strains

The yeast strains used in this study are listed in **S1 Table**. All strains are isogenic to W303 [71,72]. Yeast media and plates were prepared as previously described [73]. Strain construction for knockouts and epitope tagging was performed as described in [74]. After transformation and selection, knockouts were verified by sequencing using primers that flank each gene; western blot or fluorescent microscopy was used to confirmed epitope tagging.

### Mammalian cell culture

U2OS cells (ATCC) were cultured in DMEM w/ 10% FBS and 1x penicillin-streptomycin (Life Technologies: 50u/ml penicillin, 50 μg/ml streptomycin) at 5% $CO_2$.

### Serial Dilutions, cisplatin and mitomycin C plates

The indicated cultures were grown in 4 ml SC medium (pH 5.8) for cisplatin exposure or YPD for MMC exposure overnight at 30˚C, and subsequently diluted to 0.2 $OD_{600}$ and grown for 2.5 hours until the cultures reached logarithm phase. The cultures were five-fold serially diluted starting at 0.2 $OD_{600}$ and 5 μl were spotted on the indicated plates and grown for 2 days at 30˚C before being photographed. Photoshop was used to crop and enhance the contrast of the images, and subsequently converted to black and white images in the Figs 1–8.

Cisplatin (SelleckChem S1166) was dissolved to 100 mg/ml stock solution in dimethyl sulfoxide (DMSO). Synthetic complete (SC) plates containing the indicated doses of cisplatin, were made by diluting the stock solution. Cisplatin was handled in the dark and plates were stored in the dark and used within 24 hours.

MMC (SelleckChem S8146) was dissolved to 50 mg/ml stock solution in dimethyl sulfoxide (DMSO). YPD plates containing the indicated doses of MMC, were made by diluting the stock solution. MMC was handled in the dark and plates were stored in the dark and used within 24 hours.

### Cycloheximide chase

Cycloheximide chases in yeast were adapted from [75]. Yeast cultures were grown in 3 ml SC medium (pH 5.8) or YPD overnight at 30˚C, and subsequently diluted to 0.2 $OD_{600}$ in 35 ml SC/YPD medium and grown until the cultures reached logarithm phase. The translation inhibitor, cycloheximide (Sigma-Aldrich), was then added to a final concentration of 0.5 mg/

ml. In addition to cycloheximide, DMSO (0.1%) or cisplatin (100 μg/ml) or MMC (100 μg/ml) diluted in DMSO was added to the culture and grown at 30˚C for the indicated time points. Where indicated in the text, cells were pretreated with proteasome inhibitor MG-132 (50 μM, SelleckChem S2619) for one hour prior to CHX addition. Equal amounts of cells (0.75 $OD_{600}$) were taken at each time-point, pelleted, supernatant was removed, and washed once with $_{dd}H2O$. The pellets were flash frozen on dry ice. Protein was extracted from whole cell lysates by TCA preparation as described in 51 μl of loading buffer [76]. The 13 μl of protein was run on a 10% SDS-PAGE gel and transferred to a PVDF membrane by semidry transfer (Bio-Rad) at 13V for 2 hours 15 min. Western blot using Myc antibodies were used detect Hrq1-9Myc (α-Myc, mouse monoclonal, Santa Cruz Biotechnology, 1:500) or Kar2 (α-Kar2, rabbit polyclonal, Santa Cruz biotechnology, 1:5000) as a loading control.

Mammalian CHX chases were adapted from the yeast protocol and performed using U2OS cells. Cells were seeded into 6-well plates at 75,000 cells per well and were grown for three days in 10% FBS-DMEM. The cells were then treated with CHX (50 μg/ml) and untreated, or 50 μM cisplatin (diluted in water), or 75 μM acetaldehyde (Sigma-Aldrich). One well of each 6-well dish per time-point was collected and lysed in RIPA buffer [150 mM NaCl, 50 mM Tris-HCL pH 7.2, 1% Triton X-100, 0.5 Na-deoxycholate, 0.1% SDS, 1mM EDTA, 1x Pierce Protease Inhibitor (Thermo), 1x Phosphatase Inhibitor (Thermo)]. Protein concentration was measured via Bradford assay, equal amount of protein were run on a 10% SDS-PAGE gel, transferred to PVDF membrane, and blotted endogenous RECQL4 (rabbit polyclonal, Cell Signaling, 1:500, WB) and γ-tubulin ((mouse monoclonal, Sigma Aldrich, 1:5000) or GAPDH (mouse monoclonal, UBP Bio, 1:1000) as a loading control. The blots were scanned, the images were cropped and grayscale using Photoshop. ImageJ was used to determine relative protein quantity by mean gray value measurement. It was then normalized to the loading control and to timepoint zero.

## Cell cycle analysis

Cells expressing Hrq1-9myc were grown overnight in 4 ml YPD at RT, and subsequently diluted to 0.2 $OD_{600}$ in 30 ml YPD and grown to logarithmic phase at RT. Alpha factor (Genescript) diluted in DMSO was added to a final concentration of 10 μM, and cells were arrested in G1 for 3 hours. The asynchronous control sample was removed directly prior to alpha factor addition. To release the cells from alpha factor, the yeast was pelleted, washed twice in an equal volume of water, and transferred into 30 mL YPD medium. Samples of equal amounts of cells were taken every 20-minutes for FACS and protein analysis. Protein samples were prepared as described above, with the exception that Clb2 (G2/M cyclin) was also analyzed (rabbit polyclonal, Santa Cruz Biotechnology, 1:2000). FACS samples were pelleted and washed once with water before being fixed with 70% EtOH and stored in the dark at 4˚C. Subsequently, the samples were treated with protease and RNase and DNA was stained with propidium iodide and analyzed by FACS as described [77].

## Canavanine mutagenesis assay

Mutagenesis assay protocol was adapted from Godin *et al.* 2016 [78]. Two individual colonies of wild-type (WT), Hrq1-7KR, Hrq1-OE were grown in 3 ml of SC with 2% galactose, to induce Hrq1-overexpression, either with or without cisplatin overnight at 30˚C to saturation. Subsequently, the culture was diluted to 2 $OD_{600}$ and 200 μl was plated onto SC-ARG+CAN plate or further diluted 100,000-fold and 200 μl was plated onto SC plate (these plates were made with galactose). The plates were incubated for 3 days at 30˚C. Plates were scanned and colonies were counted using OpenCFU software [79]. Mutation rate was calculated by

counting colonies that grew on SC-ARG+CAN vs total colonies that grew on SC. For each trial, a mean rate was calculated and graphed alongside the median from multiple experiments. Mann-Whitney test was used to determined significance.

## Direct repeat recombination assays

Direct repeat recombination assays were performed as described [80] except that the cultures were grown overnight in SC with 2% galactose either with or without cisplatin. In this assay, two *leu2* heteroalleles are disrupted with EcoRI and BsteII, respectively, with an intervening *URA3* gene. Total recombination is measure by restoration of a function *LEU2* gene, which allows yeast to grow on plates that lack leucine. The experiment was done in nonuplicate and repeated. Recombination rate was calculated and graphed alongside the median from the two experiments. Mann-Whitney test was used to determined significance.

## Cell proliferation (MTS) assay

Cells were seeded at 50,000/well in a 6 well plate. The next day 30 nM of siRNAs were transfected using INTERFERin (VWR #89129–930) per manufacturer's instructions. After 48 hours, the cells were then seeded 5,000 cells/well in a 96-well plate, the remaining cells were used for western to confirm protein knockdown levels. The next morning, cells were treated with the indicated concentration of cisplatin. After 72 hours of treatment, 20 ul of MTS reagent was added to the media and allowed to incubate in the dark for two hours. Growth was measured by absorbance at 490 nm.

## Clonogenic survival assay

Initial seeding and siRNAs transfection were performed as described for the MTS assays, except that after the 48 hours of siRNA treatment, the cells were then seeded 250 cells/well in a 6-well plate. The next morning, cells were treated with the indicated concentration of cisplatin. After 24 hours of treatment, cells were washed with PBS and fresh medium was added. Colonies were grown for 17 days. For staining, cells were washed twice with PBS before being fixed with methanol for 20 minutes and stained with crystal violet solution (0.5% crystal violet, 20% methanol) for 30 minutes. Colonies were counted using ImageJ plugin, ColonyArea [81]. Each condition was plated in triplicate and the experiment was independently performed three times. Results are plotted as a normalized area, where in the treated conditions were normalized to the untreated conditions.

## Immunofluorescence

U2OS cells were seeded at 50,000 cells/well in a 6-well dish that contained a coverslip in each well. The next day, the cells were transfected with an empty plasmid, plasmid expressing RECQL4 or RECQL4-K508A under CMV promoter. 72 hours following transfection, the cells were treated with or without cisplatin for one hour in the dark. Afterwards, the cells were washed with PBS and fresh medium was added. The treated cells were allowed to recover for 2 hours. The coverslip was then removed and fixed with 4% PFA for 20 minutes in the dark at 4°C. After fixation, the sample was treated with extraction buffer and then washed and incubated overnight with α-RAD51 (1:2000). Subsequently, the samples were washed and then incubated with Goat anti-Rabbit IgG Alexa Fluor 488 (1:2000) for an hour at RT. Following secondary antibody incubation, the samples were washed then mounted onto a slide using Prolong Gold solution with DAPI. The slides were allowed to dry overnight before imaging. Foci were image with Nikon-TiE, RAD51 foci was quantified using ImageJ plug-in, Fiji. Fiji

allows for automatic non-biased analysis; the same threshold was used for all the conditions in each experiment.

## Neutral comet assay to detect ssDNA

Initial seeding and transfection and cisplatin treatment are the same as described for the immunofluorescence experiments with the exception that a coverslip was not added. Following the 2 hours recovery, the cells were trypsinized and washed before being resuspended in PBS at 100,000 cells per milliliter. Subsequently, the cells were combined with molten LMA-garose (Cat# 4250-050-02) at 1:10 (cells: agarose) ratio, and immediately 30 μl was pipetted onto two 20-well CometSlide (Cat# 4252-02K-01). The mixture solidified at 4°C before being immersed in chilled lysis solution (Cat# 4250-050-01) overnight. Subsequently, the slides were washed three times with 1x S1 nuclease buffer (50 mM NaCl, 30 mM sodium acetate, pH 4.6 and 5% glycerol). S1 nuclease (ThermoFisher #18001016) at 20 U/ml in S1 nuclease buffer and 50 mM NaCl was added to one of the two slides and incubated for 30 min at 37°C. As a control, the other slide was incubated with the same solution, but without S1 addition. The slides were then washed three times in chilled neutral electrophoresis buffer (100 mM Tris base, 300 mM sodium acetate). Afterwards, the slides equilibrated in the neutral electrophoresis buffer for 30 minutes at 4°C. The slides were then placed in a chilled electrophoresis unit (Cat# 4250-05-ES) and submerged with chilled electrophoresis buffer. Electrophoresis was then run at 21 volts for 45 minutes at room temperature (RT). After electrophoresis, the slides were transferred to DNA precipitation buffer (6.7 ml of 7.5 M ammonium acetate and 43.3 ml 95% EtOH) and incubated at RT for 30 minutes. Following DNA precipitation, the slides were immersed in 70% EtOH and incubated for 30 minutes at RT. The slides were then allowed to dry overnight at RT and subsequently stained using SYBR Gold solution (1 ul of SYBR in 30 ml Tris-EDTA buffer, pH 7.5) for 30 minutes at RT. Afterwards, excess SYBR solution was removed, and the slides were dried overnight at RT. Comets were imaged with Nikon-TiE and quantified using Comet Assay IV software by Instem.

## Bioinformatic analysis

R version 3.6.1. was used for all the bioinformatic analysis. *RECQL4* expression in different samples was determined by first transcriptomic data from TCGA and METABRIC. TCGA data were downloaded from GSE62944 and Log2 (TPM+1) values were used for downstream analysis. For the METABRIC data set, normalized probe intensity values were obtained from Synapse (Syn1688369). For genes with multiple probes, probes with the highest inter-quartile range (IQR) were selected to represent the gene.

Tumor mutation burden (TMB) calculation was performed as previous described. Briefly, TCGA mutation annotation files from 982 patients were downloaded from FireBrowse and mutation subtypes were summarized using "maftools" package_ENREF_113. Mutations subtypes were classified into truncated (nonsense, frame-shift deletion, frame-shift insertion, splice-site) and non-truncated mutations (missense, in-frame deletion, in-frame insertion, nonstop). TMB was calculated as 2X Truncating mutation numbers + non-truncating mutation numbers.

Pathway enrichment was performed by gene set variation analysis (GSVA). Using the GSVA program, we inputted the TNBC transcriptomic data from TCGA and METABRIC, and we used the defined 50 Hallmark gene sets from molecular signature database (MSigDB) as our gene sets. The program then provided which pathways and genes were enriched, we then stratified the results based on *RECQL4* expression. Clinical and pathological outcome was based on analysis from a study where cisplatin was given as a neoadjuvant therapy in 24 TNBC

tumors [64]. Transcriptomic data were downloaded from GSE18864. Clinical response to cisplatin was measured as progressive disease (PD), stable disease (SD), partial response (cPR), complete response (cCR). For clinical response, good and poor responders were defined as patients with cPR/cCR status and PD/SD status respectively. Pathological response to cisplatin was monitored by Miller Payne metric score (grade 1 is no significant tumor reduction and grade 5 is compete tumor reduction). Good and poor responders were defined as patients with a Miller Payne Score of 4–5 or 0–3 respectively.

## Antibodies

The following primary and secondary antibodies were used at the indicated dilutions: Myc (mouse monoclonal, Santa Cruz Biotechnology #sc-40, 1:500, WB); Clb2 (rabbit polyclonal, Santa Cruz Biotechnology #sc-9071, 1:2000, WB); Kar2 (rabbit polyclonal, Santa Cruz Biotechnology #sc-33630, 1:5000, WB); RECQL4 (rabbit polyclonal, Cell Signaling #2814, 1:500, WB), γ- tubulin (mouse monoclonal, Sigma Aldrich #T5326, 1:5000, WB); GAPDH (mouse monoclonal, UBP Bio #Y1040, 1:1000, WB); REV1 (mouse monoclonal, Santa Cruz Biotechnology #sc-393022); RAD51 (rabbit polyclonal, Abcam #ab63801, 1:2000, IF); Anti-Mouse HRP (Jackson ImmunoResearch Laboratories #115-035-003, 1:10000, WB); Anti-Rabbit (Jackson ImmunoResearch Laboratories #111-035-003, 1:10000, WB), Goat anti-Rabbit IgG Alexa Fluor 488 (ThermoFisher #A-11034, 1:2000, IF).

## Plasmid and oligonucleotides

The plasmids and oligonucleotides used in this study is listed in **Sheets A and B in S2 Table** respectively.

## Supporting information

**S1 Fig. 9-Myc tagging Hrq1 does not result in cisplatin sensitivity.** (A) Hrq1-9myc cells displayed similar cisplatin sensitivity compared to WT cells. The indicated yeast strains were five-fold serial diluted onto SC medium containing DMSO and/or SC medium containing the indicated amount of cisplatin. The plates were photographed after 2 days of incubation at 30˚C in the dark. (B) Hrq1 protein level is unchanged with HU and MMS treatment. Hrq1-9myc cells were grown to log phase and then treated with indicated agents for two hours before being collected for TCA prep. Cisplatin was used at 100 μg/ml (Raw densitometry data in **S11 Data**). (TIF)

**S2 Fig. Hrq1 protein levels are mildly stabilized in pol-30 ubiquitylation mutant or in the absence the E2 Ub-conjugating enzyme, *UBC13*, or the E2 Ub-conjugating enzyme, *RAD6*.** (A) Hrq1-9xMYC expressing *ubc13Δ* or *rad6Δ* cells were incubated with cycloheximide in the presence or absence of 100 μg/ml cisplatin and/or 0.1% DMSO. Protein extracts were analyzed by western blot for Hrq1 protein levels (α-MYC) or a loading control, Kar2 (α-Kar2), at the indicated time points. (B) Quantification of the proportion of Hrq1 remaining relative to time 0 (before CHX addition) and the loading control, Kar2, are plotted on the graph in log scale from *ubc13Δ* or *rad6Δ* cells. Each experiment was performed in triplicate with standard error plotted (Raw densitometry data in Sheets I-N in S1 Data). WT cisplatin treated time course is replotted from Fig 1C, for direct comparison to *ubc13Δ* or *rad6Δ* cisplatin treated cells. It is important to note that Hrq1 and the loading control, Kar2, were analyzed from the same gels to account for pipetting errors. Since Hrq1 is not as abundant as the loading control, there is a limitation for the densitometry analysis. (C) Cycloheximide chase was performed as indicated in (A) however in a *rad6Δ ubc13Δ* background (Raw densitometry data in Sheets O-Q in S1

Data).
(TIF)

**S3 Fig. Hrq1 function and regulation in ICL repair is distinct from intrastrand crosslink repair.** (A) Hrq1 functions in a different pathway as Rev1 and Ubc13 to repair ICL. The indicated yeast strains were five-fold serial diluted onto YPD medium containing DMSO and/or YPD medium containing the indicated amount of MMC. The plates were photographed after 2 days of incubation at 30˚C in the dark. (B) Hrq1 protein levels does not decreased upon 100 μg/ml MMC treatment. Exponentially growing cells with Hrq1-9xMYC were incubated with cycloheximide in the presence or absence of 100 μg/ml MMC and/or 0.1% DMSO. Protein extracts were analyzed by western blot for Hrq1 protein levels (α-MYC) or a loading control, Kar2 (α-Kar2) at the indicated time points. Quantification from three separate experiment is shown, with mean and SEM graphed (Raw densitometry data in Sheets S-U in **S1 Data**). (C) Hrq1 functions in the same pathway as Rad5 and Ubc13 to repair intrastrand adducts. The indicated yeast strains were five-fold serial diluted onto YPD medium before being treated with the indicated dosages of UV-C. The plates were photographed after 2 days of incubation at 30˚C in the dark. (D) Hrq1 functions in different pathway as Rev1 to repair intrastrand adducts. The indicated yeast strains were five-fold serial diluted onto YPD medium before being treated with the indicated dosage of UV-C. The plates were photographed after 2 days of incubation at 30˚C in the dark. (E) Hrq1 protein levels decreased following UV-C treatment. Exponentially growing Hrq1-9xMYC cells were incubated with cycloheximide then treated with indicated dosage of UV-C. Protein was extracted similarly to (B). Quantification is from three separate experiments with mean and SEM graphed (Raw densitometry data in Sheets V-X in **S1 Data**).
(TIF)

**S4 Fig. High level of Hrq1-7KR results in cell lethality.** (A) Hrq1-7KR protein levels are similar to WT in basal conditions. The indicative strains were grown overnight in 2% galactose, subsequently TCA was performed and western was run to determined protein level. (B) Stabilization or overexpression of Hrq1 does not lead to increased cisplatin sensitivity. The indicated yeast strains were five-fold serial diluted onto SC medium containing DMSO and/or SC medium containing the indicated amount of cisplatin. The plates were photographed after 2 days of incubation at 30˚C in the dark. Plates that were used with *Hrq1-OE* (GAL-HRQ1, galactose inducible/dextrose repressible promoter) strains were made with galactose instead of glucose medium. (C) Overexpression of the Hrq1-7KR mutant results in cell lethality. The indicated yeast strains were grown overnight in SC medium containing raffinose. Subsequently they were five-fold serial diluted onto galactose SC plates. The plates were photographed after 2 days of incubation at 30˚C in the dark. (D) pCMV-RECQL4-KA promotes similar overexpression compared to pCMV-RECQL4. Protein extracts from cells 3 days posttransfection. Endogenous RECQL4 was detected using α-RECQL4, GAPDH served as a control.
(TIF)

**S1 Table. List of yeast strains used in this study.**
(XLSX)

**S2 Table. List of plasmids and oligonucleotides used in this study.**
(XLSX)

**S1 Data. Densitometry for yeast cycloheximide chase experiments.**
(XLSX)

**S2 Data. Densitometry for Hrq1 cell cycle experiments.**
(XLSX)

**S3 Data. Colony count for DRR assays.**
(XLS)

**S4 Data. Colony count for DRR assays.**
(XLS)

**S5 Data. Colony count for CAN mutagenesis assay.**
(XLSX)

**S6 Data. Densitometry for mammalian cycloheximide chase experiments.**
(XLSX)

**S7 Data. Unprocessed data for MTS assay.**
(XLSX)

**S8 Data. Colony quantification for clonogenic survival assay.**
(XLSX)

**S9 Data. Unprocessed RAD51 foci count.**
(XLSX)

**S10 Data. Tail moment data from neutral comet assay.**
(XLSX)

**S11 Data. Densitometry following exposure of different dosages of DNA damaging agents.**
(XLSX)

## Author Contributions

**Conceptualization:** Thong T. Luong, Kara A. Bernstein.

**Data curation:** Thong T. Luong.

**Formal analysis:** Thong T. Luong, Zheqi Li, Nolan Priedigkeit, Kara A. Bernstein.

**Funding acquisition:** Kara A. Bernstein.

**Investigation:** Thong T. Luong, Zheqi Li, Phoebe S. Parker, Stefanie Böhm, Kyle Rapchak, Adrian V. Lee.

**Methodology:** Thong T. Luong, Kara A. Bernstein.

**Visualization:** Thong T. Luong, Zheqi Li.

**Writing – original draft:** Thong T. Luong, Kara A. Bernstein.

**Writing – review & editing:** Thong T. Luong, Kara A. Bernstein.

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
