## [Decision Letter · Decision Letter 0]

28 Mar 2022

Dear Dr Bernstein,

Thank you very much for submitting your Research Article entitled 'Hrq1/RECQL4 regulation is critical for preventing aberrant recombination during DNA intrastrand crosslink repair and is upregulated in breast cancer.' to PLOS Genetics.

The manuscript was fully evaluated at the editorial level and by independent peer reviewers. The reviewers appreciated the attention to an important problem, but raised some substantial concerns about the current manuscript. Based on the reviews, we will not be able to accept this version of the manuscript, but we would be willing to review a much-revised version. We cannot, of course, promise publication at that time.

If you decide to revise the manuscript for further consideration at PLOS Genetics, please aim to resubmit within the next 60 days, unless it will take extra time to address the concerns of the reviewers, in which case we would appreciate an expected resubmission date by email to plosgenetics@plos.org.

[LINK]

We are sorry that we cannot be more positive about your manuscript at this stage. Please do not hesitate to contact us if you have any concerns or questions.

Yours sincerely,

Dmitry A. Gordenin, Ph.D.

Associate Editor

PLOS Genetics

Gregory P. Copenhaver

Editor-in-Chief

PLOS Genetics

Reviewer's Responses to Questions

**Comments to the Authors:**

Reviewer #1: In this paper, authors examined cisplatin sensitivity of Hrq1 deleted cells in many different genetic background and suggested that Hrq1 functions in the error-free template switching pathway to mediate DNA intrastrand crosslink repair. They also showed that the stability of Hrq1 & RecQL4 decreased in cisplatin treated cells in a manner dependent upon ubiquitination & proteasome, and preventing ubiquitination of Hrq1 increased cisplatin induced recombination and mutations. Finally, they analyzed the cancer genome atlas database, and suggested that RECQL4 overexpression correlates with increased recombination & mutations.

Although this manuscript includes a few interesting observations such as increase in instability of Hrq1 & RecQL4 after cisplatin treatment, it appeared to be too preliminary for publication. Important questions have not been addressed, such as how and why Hrq1 ubiquitination increases in cisplatin treated cells and how recombination & mutations increase when Hrq1 increases. Many of the results in figures require more experiments and control groups to fully support the conclusion. In addition, descriptions in results and discussion seems to be insufficient in many parts.

I included a few from many concerns about the results in this manuscript.

In Fig. 1 c, d; Fig.2 a, b; Sup. Fig. 2

I assumed that authors include representative images in figures. However, many images in figures are overexposed and did not match well with graphs. Based on description in Materials and Methods, authors seemed to use densitometry analysis of scanned Western blots images for quantitation. Especially, in this case, images should not be overexposed.

In Fig. 1b.

You used only single dose of chemicals in this experiment. Did you use enough and comparable dose of other chemicals in this experiment? I even cannot find what dose of chemicals and irradiation you have used.

In Fig. 1e and Sup Fig. 1b

Patterns of Hrq1 ubiquitination are different in these reverse IP experiments. Smeared pattern of Hrq1 was found in Ub pulldown, but Hrq1 pull down made discrete single band. Why?

In Fig. 2a

Hrq1 appeared to be already unstable in Rad16 deleted cells even in the absence of cisplatin. Why? Again, I do not see much difference in Hrq1 stability of wt and rad16 deleted cells in these images, while graphs shows differently.

In Fig. 3a

Alpha factor arrested cells are not real G1 cells. Alpha factor arrests cell cycle before start of cell cycle at G1, and expression of many genes involved in cell proliferation can be reduced in alpha factor arrested cells. You can analyze the G1 cells in the next cell cycle or you can use other methods for cell synchronization such as nocodazole arrest and release to make sure that changes in the level of proteins indeed occur in G1.

In Fig. 3f

Differences in cisplatin sensitivity between rev1 deletion and hrq1 and rev1 deletion are so marginal, and only found in a single plate (20 ug/ml cisplatin). I am very skeptical to make any conclusion based on these observations.

In Sup Fig. 3b

Again, you have to use comparable dose of MMC in this experiment. Based on sensitivities of yeast cells to cisplatin and MMC in Fig. 3 and Sup. Fig. 3, yeast cells are much more sensitive to cisplatin than MMC at the same concentration. Therefore, the result obtained by using the same concentration of MMC is not conclusive.

In Figure 4c

The levels of Hrq1 in WCE are different. If you want to directly compare the intensities of ubiquitinated proteins in IP materials, you have to use the same amount of extracts containing the same amount of target proteins.

In Figure 5c

You need to show the cisplatin sensitivity of double knock-down of RecQL4 and any one functioning in error-free PRR pathway to suggest the role of RecQL4 in the error-free PRR.

In Fig. 6.

Rad51 staining is poor, and cisplatin treatment did not increase Rad51 foci in immunostaining and tail moments in comet assay even in U2OS cells. Therefore, I do not think that these assays are reliable in this condition.

Reviewer #2: Manuscript Title: Hrq1/RECQL4 regulation is critical for preventing aberrant recombination during DNA intrastrand crosslink repair and is upregulated in breast cancer

Authors: Thong T. Luong, Zheqi Li, Nolan Priedigkeit, Phoebe S. Parker, Stefanie Böhm, Kyle Rapchak, Adrian V. Lee, Kara A. Bernstein

This manuscript will be of interest to researchers studying DNA damage repair and tolerance, especially via homologous recombination. The study follows 3 publications which reached conflicting conclusions as to the role of the RECQ family helicase Hrq1/RECQL4 in post-replicative repair of intra-strand crosslinks by homologous recombination (Groocock et al. 2012 Mol. Cell Biol., Choi et al. 2014 J Microbiol., Röhrig et al. 2018 New Phytol.). The authors use genetic, cell biological, and cytological methods to demonstrate that Hrq1 is involved in “error free” post-replicative repair in response to cisplatin in budding yeast, and validate their results using UV-C. The Hrq1-7KR mutant, which cannot be ubiquitylated, is used convincingly to show that Hrq1 is degraded in response to cisplatin, and that loss of this regulation leads to increased mutagenesis. Genetic analysis strongly suggests this degradation depends on Rad16. RECQL4 is similarly degraded in response to cisplatin or acetaldehyde, both of which can lead to intra-strand crosslinks. MTS and clonogenic survival assays again show that the RECQL4 REV1 double knockdown (RECQL4 knockout is lethal) by siRNA leads to greater cell death than either single knockdown alone. While these results are predicted from the budding yeast studies (wherein HRQ1 knockout is viable) since REV1 is a key component of the “error prone” sub-pathway of post-replicative repair, a critical siRNA control, the back complementation with siRNA-resistant plasmids that encode these factors, is missing, and should be added to show that the effect is specific to RECQL4 and REV1. Intriguingly, RECQL4 overexpression leads to either increased RAD51 foci or their accumulation. This effect is at least partially dependent on RECQL4’s helicase activity, since overexpression of the RECQL4-KA mutant results in significantly fewer RAD51 foci relative to overexpression of the wild-type allele. Finally, bioinformatic analysis of human breast cancers show that these tumors express more RECQL4 than tumor adjacent healthy tissue. Furthermore, ER- breast cancers, which are routinely treated with cisplatin, express significantly more RECQL4 than ER+ breast cancers, which are usually treated with hormone therapy. Retroanalysis of a small group of patients with triple-negative breast cancer treated with cisplatin showed that clinical outcome is better for patients with high RECQL4 expression versus low. These bioinformatic analyses enhance the significance of the study.

Note that this manuscript does not comply with PLoS Genetics submission formatting guidelines, including the order in which the Methods section should appear in the manuscript, and the requirement for line numbers.

Major Comments:

1) Figure 3A: Whereas it is obvious from the Western blot images provided that Clb2 peaks at 60 minutes, the same is not true for the protein of interest, Hrq1-9-Myc. Instead, Hrq1-9-Myc appears to plateau at 40 minutes. To strengthen the conclusion that Hrq1 protein levels peak at 40 minutes, I recommend quantification of the results in these blots as in Figures 1C and 2A.

Between 0 and 40 minutes, the data for Hrq1-9-Myc suggest that Hrq1 is upregulated and then plateaus. For Hrq1 levels to drop again during the next cell cycle, one would expect degradation. It would be helpful to see this Western blot data extended to additional time points into the second cell cycle.

Furthermore, given that the figure legends for Figures 1C and 2A clearly state that the experiment has been replicated, the fact that this information is missing from the legend in Figure 3A leads one to believe that these experiments have not been repeated. Thus, to make this claim that Hrq1 levels peak during S/G2, I believe that the data needs to quantified, repeated, and extended to later time points, such that the return to G1 levels is clear.

2) Figure 4D: This figure uses a recombination reporter assay previously published in Godin et al. 2013 NAR. Godin et al. contains a good schematic of this assay. A similar schematic should be provided in this figure.

Based on the diagram in Godin et al., it is clear that Leu+ colonies in this system can result from repair either through homologous recombination or single-strand annealing. These two possibilities can be distinguished by determining whether the resulting colonies are Ura+ or Ura-. The implications of these two outcomes for this study should be presented in the text. Moreover, the results of this analysis showing whether damage is repaired primarily through recombination or single-strand annealing should be included in the figure.

3) Figures 5C and D: Figures 5C and D: The manuscript does not show that the siRNA is specific to RECQL4 or REV1 by back complementation with siRNA resistant plasmids encoding RECQL4 or REV1. This is a relatively standard control for these types of experiments, and would create more confidence in the results, as it would show that the results are specific to RECQL4 and/or REV1.

4) Figure 6: A reasonable rationale for the use of U2OS cells to evaluate the effects of RECQL4 overexpression on DNA repair following cisplatin exposure is given. However, recently it has been shown by Löbrich and colleagues (Elbarky et al. 2021 PNAS) that U2OS specifically lack one of two recombination resolution sub-pathways, the ATRX-dependent double Holliday junction pathway, and predominantly use the RECQ5-dependent synthesis-dependent strand annealing (SDSA) pathway. This reliance on SDSA is not observed in HeLa and other cell lines. Thus, findings relating to homologous recombination in U2OS may not extend to other human cell lines/types, and this should be noted in the text as a potential shortcoming.

5) Figure 6A: Quantifying only the total number of RAD51 foci present at a single time point is not an appropriate method to determine whether there is hyper-recombination in a given mutant. There are many alternative explanations that could explain the results observed, such as increased DNA damage, or RAD51 persistence due to unresolved recombination intermediates or lack of repair. The conclusions in this section should be revised to reflect this.

The authors should also indicate in their quantitation whether the difference in RAD51 foci between the empty vector and the RECQL4-KA mutant is statistically significant.

6) Figure 7E: It has been reported that the majority of tumors overexpress RAD51 (e.g. Klein 2008 DNA Repair). Therefore, the implications of this finding are somewhat overstated, and should be revised.

7) Discussion: Please discuss how the findings of this study fit with other studies in the field, especially Lu et al. 2017 Nat Commun., since the conclusions differ from those in this study.

Additional Comments:

Overall, the manuscript is well-written with clear figures. It would benefit from these additional revisions.

1) Methods: Under the Bioinformatic Analysis sub-section, Silver et al. 2010 is cited, which I believe refers to Silver et al. 2010 J Clin Oncol., as “(Silver et al. 2010)”. This citation style is inconsistent with the rest of the manuscript. This error is made again in the Figure 7 legend. It should be corrected in both places. In addition, this reference should be added to the Bibliography at the end of the manuscript.

2) Figures 7F and G: The Methods section indicates that the analyses in these figures are based on results from a previously published study, Silver et al. 2010 (J of Clin Oncol.?). The study should be cited when referring to the findings in Figures 7F and G in the text to properly acknowledge it. It should also be noted that this study has a relatively small sample size (n = 24).

3) Discussion: Throughout the Discussion, the manuscript refers to Figure 7 when I believe it intends to reference Figure 8.

4) Discussion: The manuscript states, “Furthermore, we find that Hrq1 is regulated by the UPS during instrastrand crosslink repair and that misregulation of Hrq1 protein levels results in increased recombination and mutagenesis. We extend this analysis to mammalian cells and show that like yeast, human RECQL4 protein levels are depleted in response to DNA crosslinks… Together our studies demonstrate a conserved role for the RECQL4 protein family in crosslink repair whose levels must be fine-tuned to prevent excess recombination and mutations.” See comments on Figures 4 and 6 above. I believe the data supporting these claims is relatively weak, and that this section should be revised to include more caveats and alternative explanations.

5) Discussion: “Our studies here demonstrate that like RECQL4, Hrq1 also plays a key role during replication and is cell cycle regulated, peaking in the S/G2 cell cycle phase.” It would be helpful if the authors could cite the original publications corresponding to the RECQL4 data and their own figures for the Hrq1 data.

Reviewer #3: In the manuscript “Hrq1/RECQL4 regulation is critical for preventing aberrant recombination during DNA intrastrand crosslink repair and is upregulated in breast cancer”, the authors provide evidence that yeast Hrq1 and the human homolog RECQL4 function to promote error-free bypass of DNA lesions through an HR directed pathway such as template switching. This conclusion is based on the findings that: 1)Hrq1 and RECQL4 are required for resistance to cisplatin and acetaldehyde (both of which can generate DNA interstrand cross-links), 2) Hrq1 functions downstream of Rad5- and Ubc13-dependent PCNA polyubiquitylation, and 3) Hrq1 and RECQL4 overexpression stimulates recombination and leads to persistent Rad51 foci. Additionally, the authors identify a conserved mechanism by which Hrq1 and RECQL4 activity is regulated through proteolytic degradation following exposure to the cisplatin or acetaldehyde. They go on to show that this degradation depends on the E3 ubiquitin ligase Rad16, lysine residues on Hrq1, and the proteosome. Finally, the authors mine cancer transcriptomics data to establish a correlation between high RECQL4 expression and high efficacy of cisplatin treatment. Overall, the authors’ conclusion are generally supported by data. However the manuscript would be strengthened by the inclusion of critical experimental controls and additional experiments to test key predictions of the authors’ model for Hrq1/RECQL4 function.

Major points

1. In Figure 1e, the authors report that a pulldown of His-tagged ubiquitin results in recovery of Hrq1 and conclude that Hrq1 is ubiquitylated. However, this experiment is missing key controls to demonstrate that the pulldown is specific for ubiquitylated species. This is especially important since the authors seem to recover large amounts on non-ubiquitylated Hrq1 in the pulldown. The authors should include a negative control such as a pulldown in the absence of of Cu2+ (when His-Ub is not expressed) or pulldown with agarose beads (no Ni) to confirm a specific pulldown. Additionally, the His-Ub pulldown protocol described in the methods section is incomplete (missing wash conditions).

2. Also concerning Figure 1e: the authors could strengthen their claim that the slowly-migrating smear detected with the Hrq1correspond to ubiquitylated Hrq1 by treating the recovered proteins with a general DUB such as USP21, which would be expected to collapse the putative ubiquitylated species into an unmodified form. As it stands, this reviewer does not find the authors’ evidence that ubiquitylated Hrq1 accumulates upon cisplatin treatment very compelling.

3. In Figure 1b, how do the authors confirm that treatment with MMC, HU, IR, and MMS were sufficient to elicit a robust DNA damage response? A trivial explanation is that the doses used for the experiment presented in 1b were too low to generate levels of DNA damage sufficient to induce Hrq1 degradation. Also, the doses used in 1b seem to be missing from the figure legend and/or methods.

4. The authors propose a model in which Rad16-mediated degradation of Hrq1 constrains Hrq1 activity to allow for efficient resolution of post-replicative repair intermediates. Implicit in this model is an assumption that Hrq1 is ubiquitylated when associated with chromatin (Figure 8). Does deletion of Rad5 or Ubc13 prevent cisplatin-induced degradation of Hrq1? One might expect that blocking PCNA polyubiquitylation would prevent Hrq1 association with DNA and thereby block Hrq1 degradation.

5. Throughout the manuscript, the authors demonstrate that Hrq1 protein levels decline after treatment with cisplatin but not after treatment with MMC, MMS, IR, or HU. Cisplatin intrastrand cross-links are generally considered NER substrates while lesions formed by the other genotoxic agents are not. Is Hrq1 activity specifically directed toward NER substrates? This would be consistent with the authors’ conclusion that Hrq1functions downstream of Ubc13 upon induction of DNA damage with UVC (another NER lesion). Are Hrq1 protein levels altered by UVC or BPDE? Are ∆hrq1 cells hypersensitive to BPDE? Alternatively, cisplatin, aldehydes, and UV can all induce DNA-protein cross-links (DPCs). Is there any evidence to suggest (or exclude) the possibility that Hrq1 participates in DPC-repair?

6. The authors report that increased RECQL4 levels correlate with better tumor response to cisplatin treatment. Do cultured cells overexpressing RECQL4 (such as the U2OS cells described in Figure 6) exhibit hypersensitivity to cisplatin?

Minor points

1. The authors indicate that acetaldehyde “can produce similar lesions to cisplatin.” The cited literature (74) demonstrated that acetaldehyde can generate dG-dG intrastrand cross-links in a short single stranded oligonucleotide by cross-linking the N2 positions of dG. This lesion is distinct from the N7 cross-links formed by cisplatin. Is there any information as to how this lesion is repaired and whether this lesion forms in chromatin under the conditions used in the experiments reported here? Also, the authors may consider parsing the chemistry of the various cross-links studied in order to orient a general reader.

2. On page 24, the authors report that loss of either UBC13 or RAD6 leads to only mild stabilization of Hrq1 following cisplatin treatment and suggest that additional E2 participate in Hrq1 ubiquitylation. Could UBC13 and RAD6 function redundantly to promote Hrq1 ubiquitylation? Have the authors attempted to measure Hrq1 stability following disruption of both UBC13 and RAD6?

3. The authors state: “We find that Hrq1/RECQL4 over-expression results in genomic instability.” What is the basis for this statement, given that the authors show that RECQL4 overexpression fails to cause an increase in DSBs or ssDNA accumulation in Figure 6b.

**Have all data underlying the figures and results presented in the manuscript been provided?**

Reviewer #1: Yes

Reviewer #2: **No: **Numerical data underlying the graphs was not provided in spreadsheet form.

Reviewer #3: Yes

PLOS authors have the option to publish the peer review history of their article (what does this mean?). If published, this will include your full peer review and any attached files.

Reviewer #1: No

Reviewer #2: No

Reviewer #3: **Yes: **Dan Semlow

---

## [Decision Letter · Decision Letter 1]

21 Jul 2022

Dear Dr Bernstein,

Thank you very much for submitting your Research Article entitled 'Hrq1/RECQL4 regulation is critical for preventing aberrant recombination during DNA intrastrand crosslink repair and is upregulated in breast cancer.' to PLOS Genetics.

The manuscript was fully evaluated at the editorial level and by independent peer reviewers. The reviewers appreciated the attention to an important topic but identified some concerns that we ask you address in a revised manuscript. We therefore ask you to modify the manuscript according to the review recommendations. Your revisions should address the specific points made by each reviewer.

[LINK]

Yours sincerely,

Dmitry A. Gordenin, Ph.D.

Academic Editor

PLOS Genetics

Gregory P. Copenhaver

Editor-in-Chief

PLOS Genetics

Reviewer's Responses to Questions

**Comments to the Authors:**

Reviewer #1: Despite the revision, the results regarding ubiquitination of Hrq1 still appear to be too preliminary, and many of the points I have raised have not been addressed.

Reviewer 1:

1. In Fig. 1 c, d; Fig.2 a, b; Sup. Fig. 2

I assumed that authors include representative images in figures. However, many images in

figures are overexposed and did not match well with graphs. Based on description in

Materials and Methods, authors seemed to use densitometry analysis of scanned Western

blots images for quantitation. Especially, in this case, images should not be overexposed.

As the reviewer pointed out, the images in these figures are from three or more experiments and

are representative images. Unfortunately, since Hrq1 is not well expressed relative to the loading

control, the loading control can appear over-exposed since the images are quantitated from the

same blots. Hrq1 is expressed from its endogenous locus and promoter. We have now included

less exposed images where possible to alleviate this technical issue.

• Densitometry analysis of overexposed images is not acceptable for quantitation. Overexposed images of the Western blotting cannot represent the actual difference in protein levels. As quantitation has not been done properly, these experiments should be repeated. It is not necessary to use the same exposure time in the control blot (GAPDH or Kar2). You have to use short exposed images to avoid saturated bands. Alternatively, you can use the ChemiDoc imager detecting chemiluminescence for quantitation. Again, saturated bands should be avoided.

2. In Fig. 1b.

You used only single dose of chemicals in this experiment. Did you use enough and

comparable dose of other chemicals in this experiment? I even cannot find what dose of

chemicals and irradiation you have used.

We agree with the reviewer, and we inadvertently left off the doses analyzed, which are now

included. In addition, we have now performed additional experiments analyzing multiple doses

(New Supplemental Figure 1b).

New Supplemental Figure 1b Legend

Steady state levels of Hrq1 upon HU and MMS treatment. Hrq1-9-Myc expressing cells were

grown to early log phase and then treated with indicated DNA damaging agent for two hours (10

and 50 mM HU, 0.003% and 0.01% MMS, or 100 μg/ml cisplatin) or IR exposed (50, 100, 200

Gy) before protein analysis of whole cell extracts by western blot. GAPDH or Kar2 was used as

a loading control. Quantification of Hrq1 (anti-Myc antibody) relative to GAPDH or Kar2 is

shown below and normalized to the untreated Hrq1-9-Myc control.

• Question is whether you used enough concentration of other chemicals or not. The new supplemental figure 1b is not helpful because it shows the results using lower concentration of HU and MMS than the concentration used in Fig 1b.

3. In Fig. 1e and Sup Fig. 1b

Patterns of Hrq1 ubiquitination are different in these reverse IP experiments. Smeared

pattern of Hrq1 was found in Ub pulldown, but Hrq1 pull down made discrete single band.

Why?

We agree with the reviewer that there were differences in the ubiquitin partner depending upon

whether ubiquitin itself or Hrq1 was immunoprecipitated. An important difference between these

experiments is that the Hrq1 pull down used 25% of the yeast culture compared to the ubiquitin

pull down. Therefore, one possibility is that more Hrq1 species can be visualized. Furthermore,

the ubiquitin pulldown relies on massive over-expression of ubiquitin and perhaps that creates

differences compared to analyzing the endogenous Hrq1 protein that is not over-expressed. We

have made a note of this in the Results.

• I hardly believe that the amount of extracts makes different pattern of ubiquitinated protein precipitates. If you think different amount of extracts make a difference, this experiment should be repeated using the same amount of extracts.

5. In Fig. 3a

Alpha factor arrested cells are not real G1 cells. Alpha factor arrests cell cycle before start

of cell cycle at G1, and expression of many genes involved in cell proliferation can be

reduced in alpha factor arrested cells. You can analyze the G1 cells in the next cell cycle or

you can use other methods for cell synchronization such as nocodazole arrest and release to

make sure that changes in the level of proteins indeed occur in G1.

We respectfully disagree with the reviewer as alpha factor is extensively used to arrest cells in

G1. Regardless, our main conclusion is to demonstrate that Hrq1 expression peaks before Clb2

expression.

• You described in the manuscript that you did this experiment to see whether its protein levels are cell cycle regulated, and Hrq1 protein levels peak in S/G2. While Clb2 clearly shows a peak, Hrq1 protein level begins to increase and then reaches the plateau. Unless you show decrease in Hrq1 levels during the cell cycle, you cannot say its level peaks in S/G2.

7. In Sup Fig. 3b

Again, you have to use comparable dose of MMC in this experiment. Based on sensitivities

of yeast cells to cisplatin and MMC in Fig. 3 and Sup. Fig. 3, yeast cells are much more

sensitive to cisplatin than MMC at the same concentration. Therefore, the result obtained

by using the same concentration of MMC is not conclusive.

Unfortunately, we cannot directly compare doses of different damaging agents as they cause

different types of DNA damage and their effects not necessarily the same on hrq1Δ cells as

observed by the different sensitivity. However, we have provided multiple cisplatin and MMC

doses for analysis here (Supplemental Figure 3a).

• When you analyzed the stability of Hrq1, you used 100 ug/ml of cisplatin which is more than 3 times higher than the highest concentration used in the repair assay (30 ug/ml). As you used very high concentration of cisplatin, the effect shown here could be indirect. The concentration of MMC used in stability test (100 ug/ml) is only 1.3 times higher than the concentration used in repair assay (75 ug/ml). What would be the stability of Hrq1 if you use a higher concentration of MMC?

8. In Figure 4c

The levels of Hrq1 in WCE are different. If you want to directly compare the intensities of

ubiquitinated proteins in IP materials, you have to use the same amount of extracts

containing the same amount of target proteins.

We agree with the reviewer that the levels of Hrq1 in WCE are different either due to pipetting

error or to degradation from cisplatin exposure. To account for the difference in the WCE, we

normalized the IP to the WCE, which mitigates any of these technical differences.

• The difference between wild type and mutant pull-down is too small. Considering the many steps required for ubiquitination assay, this small difference could be simple variation between experiments. It is difficult to reach any conclusion unless a clear and significant difference is observed. In addition, you have to show the Western blot of Ub to show that a similar amount of ubiquitins were pulled-down in this experiment.

Reviewer #2: I appreciate the care with which the authors attempted to address all the reviewers’ comments in their revisions. Many of the key issues raised by the reviewers have been addressed in a satisfactory manner, improving the quality of the manuscript, and strengthening its conclusions. Several key issues remain.

Major Comments:

Lines 205-208, Figure 3A: I appreciate the inclusion of the quantitation of the Western blots for this figure. As observed from the blots and quantitation, Clb2 clearly is most strongly expressed at 60 minutes, after which its expression decreases. The same is not true for Hrq1. According to the quantitation, it appears that Hrq1 expression peaks at 80 or 100 minutes, with very little change in expression from 40 to 120 minutes. It is not clear from this analysis when Hrq1 expression will again return to the low levels observed at the beginning of the time course. The claims about this data should be softened, as they do not appear to support the conclusion that Hrq1 protein levels peak at 40 minutes.

Figures 5C and 5D: The response that Lu et al. 2017 Nat Comm has the correct siRNA control to show that expression of siRNA-resistant RECQL4 restores the wild-type phenotype is well-taken. An siRNA-resistant REV1 plasmid could still be created so that the authors can perform this important control. Several companies (e.g. Twist Biosciences) now will clone up to 5 kbp into a plasmid of your choice at a reasonable cost. However, I appreciate that the authors summed up their results in a careful way (lines 397-400).

Minor Comments:

Lines 115-117: As stated previously, analysis of RAD51 foci counts alone is not sufficient evidence to conclude that there is increased recombination. Increased RAD51 foci could also indicate failure to resolve recombination intermediates. I appreciate the authors mentioning these possibilities in lines 413-415.

Lines 345-346: To incorporate the result that both HR and SSA are increased, these lines might be rephrased to say, “These results suggest high levels of Hrq1 following cisplatin exposure results in increased homology-directed repair…” or something to this effect.

Reviewer #3: In this revised manuscript the authors have provided additional data and clarifications to support their assertion that yHrq1 and hRECQL4 promote error-free, HR-dependent bypass of lesions induced by genotoxins that general induce DNA intrastrand cross-links. They also extend their analysis of a conserved mechanism for regulating yHrq1 and hRECQL4 through proteasome-dependent degradation to show that yHrq1 is also degraded upon UVC treatment. The conclusions are generally well founded, and the authors have allayed several of my concerns. However, I still have concerns as to quality/utility of some of the data.

1) I am still unconvinced that Hrq1 undergoes ubiquitylation in response to cisplatin-induced DNA damage. While this model is certainly plausible, the His-Ub pulldown experiments (despite the authors’ efforts) sill lack appropriate specificity controls and no evidence is provided to support the conclusion that slower migrating species observed in His-Ub pulldown experiments correspond to ubiquitylated species. Additionally, the reported enhanced recovery of the ubiquitylated species is not internally controlled (instead relying on comparison with the WCE lane) and could therefore arise as a loading artifact. This is especially problematic since both the putative ubiquitylated forms and unmodified Hrq1 are recovered in these experiments and the recovery of both forms appears to be increased in response to cisplatin. Results indicating a failure to recover ubiquitylated species in the presence of the Hrq1-7KR mutated protein when pulling down Hrq1-7KR OR His-Ub (4c and S1d) are also uninformative since these experiments do not appear to have been performed in parallel with experiments using wild-type Hrq1 for comparison. Overall, I strongly recommend omitting data relating to the ubiquitylation of Hrq1 since the reported experiments are unconvincing and detract from the rest of the manuscript (which is compelling).

2) Data demonstrating that the K164R mutant stabilizes Hrq1 upon cisplatin treatment would enhance that paper. As presented in S2d (without a wild-type PCNA condition for reference), however, the effect of the K164R mutation cannot be assessed.

**Have all data underlying the figures and results presented in the manuscript been provided?**

Reviewer #1: Yes

Reviewer #2: **No: **My understanding from having published in PLoS Genetics previously and reading the data availability policy is that Excel files containing all the underlying values from the plots should be given for every figure. E.g., the policy states, "For example, authors should submit the following data:

The values behind the means, standard deviations and other measures reported;

The values used to build graphs;"

Based on these standards, it seems the authors should make an Excel file with this information with a separate tab for each figure.

Reviewer #3: Yes

PLOS authors have the option to publish the peer review history of their article (what does this mean?). If published, this will include your full peer review and any attached files.

Reviewer #1: No

Reviewer #2: No

Reviewer #3: No

---

## [Decision Letter · Decision Letter 2]

18 Aug 2022

Dear Dr Bernstein,

We are pleased to inform you that your manuscript entitled "Hrq1/RECQL4 regulation is critical for preventing aberrant recombination during DNA intrastrand crosslink repair and is upregulated in breast cancer." has been editorially accepted for publication in PLOS Genetics. Congratulations!

Please, also consider two comments of Reviewer 2 and either make changes or explain, why these changes are not necessary. This should be explained in a separate note within a submission package.

You may disregard the notice of reviewer 2 about the lack of numerical data that underlies graphs or summary statistics.  In the follow-up correspondence between Reviewer 2 and PLOS Genetics office, it was clarified that the support files that could be seen via hyperlink in the combined pdf were not accessible to the reviewer by apparently technical reasons.  AE was able to access/inspect these files and found them satisfactory.  Please, assure that all data files are included in the Supporting Information at submission. The technical team will verify that this happens within the pre-publication process.

Yours sincerely,

Dmitry A. Gordenin, Ph.D.

Academic Editor

PLOS Genetics

Gregory P. Copenhaver

Editor-in-Chief

PLOS Genetics

Comments from the reviewers (if applicable):

Reviewer's Responses to Questions

**Comments to the Authors:**

Reviewer #2: Major Comments:

Lines 196, 197: “These results demonstrate that Hrq1 is enriched during S/G2, the cell cycle stage when PRR functions.” A protein whose expression plateaus during S/G2 is not only enriched during S/G2, but also all stages except G1. The only conclusion that can be drawn is that it is highly unlikely the protein acts during G1.

Figures 5C and 5D: Without complementing with siRNA-resistant REV1, the data shown in these figures is significantly weaker.

No minor comments.

Reviewer #3: In this revised manuscript, the author's present a streamlined analysis of the role of HRQ1 in intrastrand cross-link repair. All of my concerns have been addressed.

**Have all data underlying the figures and results presented in the manuscript been provided?**

Reviewer #2: **No: **Again, PLoS Genetics policy clearly states, "numerical data that underlies graphs or summary statistics should be provided in spreadsheet form as supporting information." I do not see this data provided as a supplemental file for this manuscript.

Reviewer #3: Yes

PLOS authors have the option to publish the peer review history of their article (what does this mean?). If published, this will include your full peer review and any attached files.

Reviewer #2: No

Reviewer #3: No

**Data Deposition**

http://datadryad.org/submit?journalID=pgenetics&manu=PGENETICS-D-22-00258R2

**Press Queries**

---

## [Editor Report · Acceptance letter]

3 Sep 2022

PGENETICS-D-22-00258R2 

Hrq1/RECQL4 regulation is critical for preventing aberrant recombination during DNA intrastrand crosslink repair and is upregulated in breast cancer. 

Dear Dr Bernstein, 

We are pleased to inform you that your manuscript entitled "Hrq1/RECQL4 regulation is critical for preventing aberrant recombination during DNA intrastrand crosslink repair and is upregulated in breast cancer." has been formally accepted for publication in PLOS Genetics! Your manuscript is now with our production department and you will be notified of the publication date in due course.

With kind regards,

Olena Szabo

PLOS Genetics

On behalf of:
